# Learning Classifiers That Induce Markets

Yonatan Sommer [1]   Ivri Hikri [* 1]   Lotan Amit [* 1]   Nir Rosenfeld [1]

## Abstract

When learning is used to inform decisions about humans, such as for loans, hiring, or admissions, this can incentivize users to strategically modify their features, at a cost, to obtain positive predictions. The common assumption is that the function governing costs is exogenous, fixed, and predetermined. We challenge this assumption, and assert that costs emerge as a *result* of deploying a classifier. Our idea is simple: when users seek positive predictions, this creates demand for important features; and if features are available for purchase, then a market will form, and competition will give rise to prices. We extend the strategic classification framework to support this notion, and study learning in a setting where a classifier can induce a market for features. We present an analysis of the learning task, devise an algorithm for computing market prices, propose a differentiable learning framework, and conduct experiments to explore our novel setting and approach.

## 1. Introduction

Strategic classification (Hardt et al., 2016; Brückner et al., 2012) considers learning in a setting where users can strategically manipulate their features to obtain positive predictions. This applies to tasks such as loan approval, job hiring, school admissions, insurance claims, and welfare benefits, in which the interests of users (e.g., getting the loan or being hired) may not be aligned with the system's learning objective of maximizing accuracy. The primary goal of strategic learning is to train classifiers that are robust to such responsive user behavior, an idea that has gained much recent traction (see Sec. 1.1 for a partial list of related work).

A core assumption of strategic classification is that feature manipulation is *costly*, i.e., that modifying $x$ to some other

$x'$ incurs a cost to the user. These costs are typically modeled via a *cost function* $c(x, x')$ that underlies user decisions, and hence governs strategic behavior. The vast majority of the literature considers costs as predetermined and fixed; even if unknown to the learner, costs are still assumed to simply 'exist'. But where do costs come from, what form do they take, and how do they come to be? Challenging the conventional assumption of exogenous costs, our works sets out to propose and study alternative cost mechanisms.

One such alternative, and the focus of our paper, is the idea that costs can materialize through *market forces*: the classifier creates demand, suppliers set prices, and users pay for items or services that aid them in securing positive predictions. As an example, consider university admissions, which often rely on standardized test scores (e.g., SAT). Since these affect acceptance decisions, students are incentivized to improve their scores; this, in turn, has created a (billion-dollar) market for preparation courses. We posit that the price of such courses is determined by the importance of standardized tests as a feature in the decision rule for admission: if a policy update changes the relative weight of test scores, then prices should adjust accordingly.[1] Note that such changes also affect who will—and even who *can*—take such costly courses. This, in turn, can affect the eventual composition of admitted students. If a learned classifier is to be used to inform such decisions, then learning must be aware of, and accountable for, the market it fosters.

Our paper formalizes this idea and applies it to the framework of strategic classification. When users seek positive predictions, this creates demand for features that are important for classification; and if features are available for 'purchase' from sellers, then a competitive market is formed. The cost of obtaining features is then determined by their market price, which is reflective of their market value, and users can purchase any bundle of features whose price is within their budget. Crucially, prices are not given nor predetermined; rather, they depend on the learned classifier through how it shapes demand, as it relates to the entire data distribution. This means that to obtain a strategically robust classifier, learning must be able to anticipate the market it induces. We refer to this as *market-aware classification*. To

*Equal contribution [1]Faculty of Computer Science, Technion – Israel Institute of Technology, Haifa, Israel. Correspondence to: Nir Rosenfeld <nirr@cs.technion.ac.il>.

*Proceedings of the 42$^{nd}$ International Conference on Machine Learning*, Vancouver, Canada. PMLR 267, 2025. Copyright 2025 by the author(s).

---

[1]For broader discussion on changes in SAT policy as they relate to strategic behavior see Liu & Garg (2021).

facilitate learning in this setting, we (i) define 'a market for features', as it relates to the learning task; (ii) characterize an appropriate notion of price equilibrium; and (iii) study the task of learning strategic classifiers that induce markets.

Learning in our market setting admits two key challenges. First, since market prices rely on the aggregate demand of *all* users, the behavior of users becomes dependent through the market mechanism. This is in sharp contrast to the standard setting in which users respond independently and the objective decomposes over training examples. As we show, this can have a stark effect on learning, since even points that lie far from the decision boundary can still have a significant impact on market prices, and hence on the behavior, of others. Fortunately, a useful property of equilibrium prices is that if the market is efficient, then prices reflect all relevant information. In our setting, this implies that *conditioned on prices*, the objective does decompose over users. This allows us to adapt standard techniques for strategic learning with (independent) user responses to handle (dependent) market-induced cost functions. Our second challenge is therefore to compute market prices effectively and as part of the training pipeline. For this, we first give an algorithm for computing prices exactly, and show that it is efficient. We then propose a differentiable variant of the algorithm that computes 'smooth' prices, which enables end-to-end optimization of the entire objective using gradient methods.

Using our approach, and via both synthetic and semi-synthetic experiments, we proceed to explore the effects of induced markets on learning and its outcomes, Our main results here are twofold. First, we show that markets can give rise to complex behavioral patterns that differ significantly from the conventional strategic setting. Under a fixed cost function (e.g., some norm), whether a point will move or not depends only on its distance from the decision boundary. Conversely, since market prices are *adaptive*, the cost-effectiveness of moving for any given point depends on the aggregate demand induced by the particular classifier. This means that which points move and which don't can follow highly irregular patterns, and in some cases counterintuitive. One example is that 'raising the bar' on acceptance by increasing the threshold—a common approach to combat strategic behavior—can cause *more* points to cross. Another surprising outcome is that unseparable data can *become* separable. This can occur when (i) budgets correlate with labels, and (ii) prices discriminate against low-budget users.

This drives our second result, which is that budgets play a distinctive role in shaping learning outcomes. Our framework makes a distinction between what users have (i.e., their features) and their economic stature (via their budget). Here we show that learning tends to favor users with larger budgets. The mechanism for this is indirect: if the classifier separates the data well but in a way that negative points

hold most of the aggregate budget, then prices will be low, negative points will cross the decision boundary—and accuracy will be reduced. Thus, since classifiers gain power over the induced market, maximizing accuracy will often be achieved by learning classifiers under which positive predictions are indirectly associated with high budgets. This raises natural questions regarding socioeconomic equity—an important yet underexplored notion of fairness.

## 1.1. Related work

**Strategic classification.** The field of strategic classification (Brückner et al., 2012; Hardt et al., 2016) has gained much recent interest. This has led to many advances both in theory (Sundaram et al., 2021; Zhang & Conitzer, 2021) and in practice (Levanon & Rosenfeld, 2021). The original formulation includes several strong assumptions, in particular regarding costs, which subsequent works have challenged or relaxed. One line of research considers learning under unknown (but nonetheless fixed) costs, including in the online (Dong et al., 2018; Ahmadi et al., 2021), multi-round batch (Lechner et al., 2023), and one-shot (Rosenfeld & Rosenfeld, 2024) settings; under personalized costs (Lechner et al., 2023; Shao et al., 2024) and for general manipulation graphs (Ahmadi et al., 2023; Cohen et al., 2024). A related thread relaxes assumptions on user-side information, but focuses on uncertainty regarding the classifier rather than costs (Ghalme et al., 2021; Bechavod et al., 2022; Barsotti et al., 2022). Another assumption is that users respond independently; this has been relaxed by injecting dependencies through the utility function in a ranking task (Liu et al., 2022), or through the model class by making use of a network structure over users (Eilat et al., 2023). In a recent work, Hossain et al. (2025) augment the cost function to include externalities, which entail dependencies. Our work proposes that user behavior becomes dependent through a market mechanism in which demand, and therefore prices, derive from the classifier.

**Learning and markets.** A large literature considers markets for data (e.g., Agarwal et al., 2019; Ghorbani & Zou, 2019; Chen et al., 2022) or trained models (e.g., Chen et al., 2019; Huang et al., 2023). A recent line of work studies competition between platforms or service providers (Ben-Porat & Tennenholtz, 2017; 2019; Guo et al., 2022; Jagadeesan et al., 2023; 2024; Shekhtman & Dean, 2024; Einav & Rosenfeld, 2025). Here learning is used to elicit user preferences, e.g. towards making useful recommendations (Hron et al., 2023; Eilat & Rosenfeld, 2023). In contrast, our setting considers how learning *creates* a market, where the commodity is features. The idea that features affect demand has been considered in Nahum et al. (2024), but for market decongestion via feature selection and in a different setting. Closer to ours in spirit, Hardt et al. (2022) measure the power of learning to shape outcomes through

predictions that cause a distribution shift. However, they target a general performative setting in which neither a market nor user incentives are explicitly modeled. Epasto et al. (2018) study data-driven algorithms for mechanism design (e.g., auctions) in which rational agents can misreport information (e.g., bid untruthfully). Interestingly, they conclude that learning a mechanism is possible if misreporting bears a cost to users—as in strategic classification.

## 2. Setup

**Strategic classification.** In standard strategic classification, users are described by features $x \in \mathbb{R}^d$, and have binary labels $y \in \{0, 1\}$. Given a sample set of pairs $(x, y)$ drawn iid from some unknown joint distribution $D$, the goal in learning is to find a classifier $h$ from some model class $H$ whose predictions $\hat{y} = h(x)$ obtain high expected accuracy on future samples. Our focus will be on linear classifiers, $h_{w,\tau}(x) = \text{sign}(w^\top x + \tau)$. The challenge in strategic learning is that users can 'game' the system by manipulating their features to obtain positive predictions. In particular, given the classifier $h$, users are assumed to be rational and therefore modify their features via the *best-response mapping*:

$$\Delta_h(x) = \underset{x'}{\text{argmax}} \, h(x') - c(x, x') \qquad (1)$$

where $h(x') \in \{\pm 1\}$ is their utility gained from prediction outcomes on the modified input, and $c(x, x')$ is a cost function that governs the costs of changing $x$ to any other $x'$. The goal is then to learn a classifier $h$ that is robust to such strategic responses, and the strategic learning objective is:

$$\underset{h \in H}{\text{argmin}} \, \mathbb{E}_D\left[\mathbb{1}\{y \neq h(x^h)\}\right], \quad x^h = \Delta_h(x) \qquad (2)$$

**Market setting.** Our setting builds on the above to allow for the formation of a market for features. To generally enable transactions, we require two additional structural assumptions. First, we assume features describe tangibles; this means that each $x_{[i]} \geq 0$, and that a larger value means having 'more' of feature $i$. Second, we assume each user has an (individualized) monetary budget $b \geq 0$, which limits the amount they are willing to spend (or, equivlently, the value they attribute to obtaining a positive prediction). This extends the joint distribution to be over tuples $(x, b, y) \sim D$.

Apart from these, the only distinction of our setup is that we use a particular cost function to express market costs. We will make use of *linear costs* (Hardt et al., 2016); given a vector $\boldsymbol{p} = (p_1, \ldots, p_d) \geq 0$ and for $\delta = x' - x$, define:

$$c_{\boldsymbol{p}}(x, x') = c_{\boldsymbol{p}}(\delta) = \delta^\top \boldsymbol{p} \qquad (3)$$

If we consider $\delta$ as a bundle of features, then we can interpret $\boldsymbol{p}$ as a vector of *prices*, where each $p_i$ is the price of purchasing one unit of feature $i$. We assume that users

can buy features (but cannot sell), so that $\delta \geq 0$; together with $\boldsymbol{p} \geq 0$, this ensures $c_{\boldsymbol{p}}(\delta) \geq 0$ always.[2] Rather than assuming prices are fixed and given, our main innovation is to let $\boldsymbol{p}$ be determined by forces of supply and demand.

**Sellers and market prices.** Our setting assumes there are $d$ distinct sellers, $s_1, \ldots, s_d$, where each seller $s_i$ sells feature $i$ exclusively and can determine its price $p_i$. The goal of each seller is to maximize her *expected revenue*, defined as:

$$r_i(\boldsymbol{p}) = p_i \cdot \mathbb{E}_D[\delta_i(x; \boldsymbol{p})] \qquad (4)$$

where $\delta_i$ is the amount of feature $i$ purchased by user $x$ at prices $\boldsymbol{p}$. We consider a setting of unlimited supply (e.g., as in digital goods) and in which users can purchase any real quantity of any feature, $\delta_i \in \mathbb{R}_+ \, \forall i \in [d]$.

Note revenue to seller $s_i$ depends not only on its $p_i$, which it controls, but also on all *other* prices, $p_{-i}$, which are set by other sellers. As such, we will assume that prices reach equilibrium, denoted $\boldsymbol{p}^* = (p_1^*, \ldots, p_d^*)$, which is revenue-maximizing in the sense that no seller $s_i$ can improve her own revenue by changing $p_i$, given that all other prices remain fixed. We refer to $\boldsymbol{p}^*$ as 'market prices', and will define them precisely in Sec. 3. A crucial point is that market prices depend on the joint demand for all features, aggregated over all users. This, in turn, is shaped by the choice of classifier, as we describe next.

**Classifiers that induce markets.** Since by Eq. (1) utility to users derives from their prediction $\hat{y} = h(x)$, any user $x$ who is classified as negative (i.e., has $h(x) = 0$) will be interested in purchasing additional features $\delta = (\delta_1, \ldots, \delta_d)$ if this results in flipping her prediction to $h(x + \delta) = 1$. The demand set of a user therefore includes all $\delta$ for which:

$$w^\top(x + \delta) + \tau \geq 0 \quad \text{and} \quad \delta^\top p \leq b \qquad (5)$$

Overall demand is then given by aggregating all such $\delta$ over the collection of all users, and market prices $\boldsymbol{p}^*$ are set by sellers to maximize revenue under this global demand set.

Notice how demand, and therefore prices, depend on the interaction between the data distribution (i.e., all pairs $(x, b)$) and the classifier $h$ (via $w$ and $\tau$). In this sense, we get that *each choice of classifier induces a market*. We will henceforth use $\boldsymbol{p}^h := \boldsymbol{p}^*(h; D)$ to denote the classifier-dependent equilibrium prices that govern user responses in the market.

**Strategic learning objective.** Given a sample set $S = \{(x_i, b_i, y_i)\}_{i=1}^m$ drawn iid from $D$, we will be interested in learning a classifier that maximizes expected accuracy under the market it induces. This requires us to anticipate how users will respond: for a given $h$, plugging Eq. (3) into

---

[2]Note this circumvents an artifact of linear costs, made apparent in Hardt et al. (2016), which is that points can move 'for free' in any direction (and hence any distance) that is orthogonal to $w$.

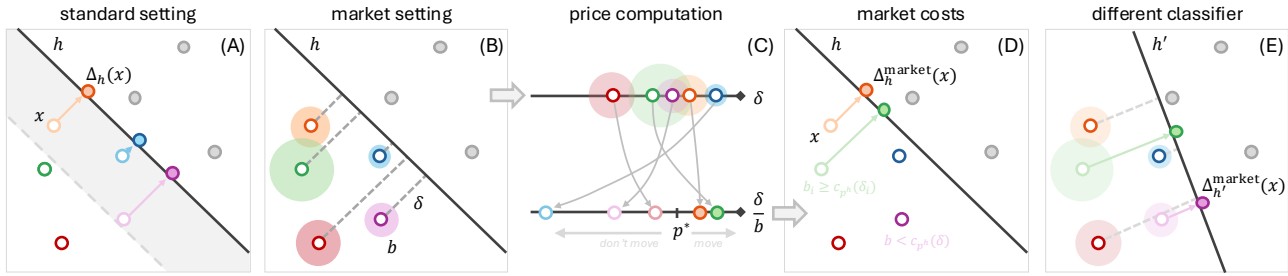

Figure 1: **Strategic classification under market costs. (A)** In the standard setting with predetermined norm costs, each classifier $h$ induces a fixed region of points that will cross (light grey). **(B)** In our market setting, points have budgets $b$ (circles), and their distance to $h$ determines their demand $\delta$ (dashed lines). **(C)** Demand for all users is projected onto a single demand space (top) and then normalized by budgets (bottom), over which revenue-maximizing prices $\boldsymbol{p}^h$ are computed. **(D)** Under market costs, points move if their budget permits buying sufficient features, and do not move otherwise. **(E)** A different classifier $h'$ creates different demand, and therefore induces different prices $\boldsymbol{p}^{h'}$. This can result in different points moving.

Eq. (1) gives the *market best-response* mapping:

$$\Delta_h^{\texttt{market}}(x,b) = x + \delta_x,$$

$$\delta_x = \operatorname*{argmax}_{\delta \geq 0} h(x + \delta) - \frac{1}{b}c_{\boldsymbol{p}^h}(\delta) \quad (6)$$

which satisfies budget constraints implicitly. Given Eq. (6) can be interpreted as each user having individualized costs $c_x(\delta) = \frac{1}{b}c_{\boldsymbol{p}^h}(\delta)$. Nonetheless, a crucial point is that when $h$ is learned, individual best-responses $\Delta_h^{\texttt{market}}(x,b)$ become dependent on all other users, since $\boldsymbol{p}^h$ depends on the distribution $D$ through the learned $h$. Thus, users respond as a collective—not independently. Fig. 1 illustrates the process in comparison to standard strategic classification.

Given Eq. (6), our strategic learning objective is:

$$\operatorname*{argmin}_{h \in H} \mathbb{E}_D\big[\mathbb{1}\{y \neq h(x^h)\}\big], \quad x^h = \Delta_h^{\texttt{market}}(x,b) \quad (7)$$

which in practice we will replace with an appropriate empirical proxy (Sec. 4). Note $h$ is a function of features alone—and not of budgets; thus, we assume budgets are observed at train time, but at test time are private to users, and affect their computation of $x^h$. This makes $\Delta_h^{\texttt{market}}$ a special case of the generalized response model proposed in Levanon & Rosenfeld (2022) which supports private information, albeit with cross-user dependencies (which are unsupported).

## 3. Market Prices: Analysis and Algorithm

Optimizing Eq. (7) requires the ability to anticipate how users respond to the market. By Eq. (6), this can be achieved by computing induced prices $\boldsymbol{p}^h$. Our first task is therefore to compute market prices for a given $h$. We begin with analyzing the market, and then give an exact pricing algorithm.

**How users respond to prices.** Given a classifier $h$ and a general price vector $\boldsymbol{p}$, how will users behave? To gain insight, we will first consider the case of $w > 0$, and later generalize. Notice that computing $x^h$ in Eq. (6) can be broken

down into three steps: First, compute $\hat{y} = h(x)$, and proceed only if $\hat{y} = -1$. If so, then second, find the least-costly $\delta$ that gives a positive prediction, this by solving the LP:[3]

$$\delta_* = \operatorname*{argmin}_{\delta \geq 0} \delta^\top \boldsymbol{p} \quad \text{s.t.} \quad w^\top(x + \delta) + \tau = 0 \quad (8)$$

Third, apply $x^h = x + \delta_*$ iff budget permits, i.e., $\delta_*^\top \boldsymbol{p} \leq b$.

To understand $\delta_*$, consider a change of variables in Eq. (8) using $z_i = \delta_i w_i$. The LP can now be rewritten as:

$$\operatorname*{argmin}_{z \geq 0} \sum_{i=1}^d \frac{p_i}{w_i} z_i \quad \text{s.t.} \quad \sum_{i=1}^d z_i = \kappa_x \quad (9)$$

where the constant $\kappa_x = -w^\top x - \tau$ is non-negative for relevant points (i.e., having $\hat{y} = -1$). This provides an alternative interpretation: the user must allocate $\kappa_x$ mass across features, using $z$, to minimize total cost-per-value, where 'values' correspond to entries of $w$. This has a simple solution, which is to set $z_i = \kappa_x$ if $i$ attains the minimal ratio $p_i/w_i$, and 0 otherwise. If multiple features attain the minimum, then these features are substitutable, and so any allocation of $z$ among them is equivalently optimal. In other words, users will purchase only the most cost-effective features, but are indifferent within this set of features.

**How prices adjust to demand.** Given the above, we next consider how sellers should set prices. Notice how by Eq. (9), all user decisions depend on the ratios $p_i/w_i$, and differ only in the constant $\kappa_x$. Since all users buy only the most cost-effective features, any $s_i$ whose ratio is *not* minimal will receive zero market share. Sellers therefore compete over who attains the minimal $p_i/w_i$. Since sellers in our setting have no capacity constraints or production costs, we assume sellers have foresight and so coordinate to prevent

---

[3]Since users minimize costs, we can use an equality constraint.

**Algorithm 1** Exact empirical market prices

1: **input:** classifier $h_{w,\tau}$, sample set $S = \{(x_i, b_i, y_i)\}_{i=1}^m$
2: **initialize:** $r = 0$ (revenue), $U = 0$ (total units sold)
3: **for** $i = 1, \ldots, m$ **do**
4:     $u_i \leftarrow \mathtt{dist}^+(x_i; h)$
5:     $\bar{u}_i \leftarrow u_i/b_i$
6: $(\bar{u}_{(1)}, \ldots, \bar{u}_{(m)}) \leftarrow \mathtt{sort}(\bar{u}_1, \ldots, \bar{u}_m)$
7: **for** $i = 1, \ldots, m$ **do**
8:     $p_i \leftarrow 1/\bar{u}_{(i)}$
9:     $U \leftarrow U + \bar{u}_{(i)}$
10:     **if** $p_i U > r$ **then**
11:        $r \leftarrow p_i U$
12:        $\hat{p} \leftarrow p_i$
13: **return:** $\hat{\boldsymbol{p}} = \hat{p} \cdot w/\|w\|$

price collapse.[4] This gives the equilibrium condition:

$$\forall i \in [d], \qquad \frac{p_i}{w_i} = \rho^* > 0 \tag{10}$$

for $\rho^*$ that admits maximal total revenue, $r = \sum_i r_i$. This implies a tight connection between the classifier and prices:

**Proposition 1.** *Let* $h(x) = \mathrm{sign}(w^\top x + \tau)$*, then there exist equilibrium market prices* $\boldsymbol{p}^*$ *that are proportional to* $w$:

$$\boldsymbol{p}^*(h; D) = \rho^* \cdot w \tag{11}$$

*for some* $\rho^* \in \mathbb{R}_+$ *which also depends on* $h$ *and* $D$.

The particular equilibrium in Eq. (11) will become highly useful for our method in Sec. 4. Interestingly, for the purpose of learning, prices can be computed as in Eq. (11) even if some entries in $w$ are negative. This is because for every $h$ there exists a price vector $\boldsymbol{p}' \geq 0$ such that (i) $\boldsymbol{p}'$ is an equilibrium price, and (ii) outcomes under $\boldsymbol{p}^*$ and $\boldsymbol{p}'$ are the same, i.e., the same set of points cross. Intuitively, this is due to items being exchangeable; see Appendix A.1.

**Computing empirical market prices.** Given a classifier $h$ and sample set $S = \{(x_i, b_i, y_i)\}_{i=1}^m$, we will be interested in computing revenue-maximizing market prices. Because we only have a sample at hand, our goal will be to compute optimal *empirical market prices* $\hat{\boldsymbol{p}}$. Applying Eq. (11) to the empirical distribution over $S$, we get that $\hat{\boldsymbol{p}} = \hat{\rho}w$ for the scalar $\hat{\rho}$ which maximizes total empirical revenue, denoted $\hat{r}$. This is highly useful, since the problem of computing equilibrium for the empirical market under a given $h$ reduces to optimizing over scalars $\rho \in \mathbb{R}$.

By Eq. (11) we can find market prices in the direction of $w$. The demand of a user is the (directional) distance from $x$ to the decision boundary of $h$, measured in 'units' $u \in \mathbb{R}_+$.

---

[4]This is essentially Bertrand's paradox, for which we invoke the folk theorem to enable the formation of cooperative equilibrium.

**Observation 1.** *Let* $h$ *and* $S$*, then for a given user* $x$*, and for any* $\rho \in \mathbb{R}$*, her demand under prices* $\boldsymbol{p} = \rho w$ *is:*

$$u = \mathtt{dist}^+(x; h) = \max\left\{0, -\frac{w^\top x + \tau}{\|w\|}\right\} \tag{12}$$

Here the max over 0 ensures that demand is considered only for relevant users, i.e., for which $h(x) = -1$. This transition to units of demand lays the ground for our algorithm.

**Exact aglorithm.** Algorithm 1 provides pseudocode for an algorithm that efficiently computes the optimal (scalar) prices $\hat{\rho}$ for given $h$ and $S$. Our key observation is that it suffices to work with *units-per-budget*, defined as $\bar{u}_i = u_i/b_i$ for each user $x_i$. The first steps are therefore to project demand onto $w$, obtain all $u_i$, and normalize by $b_i$ to get $\bar{u}_i$. Correctness of the algorithm follows from the next result:

**Theorem 1.** *Given (uni-dimensional) demand* $\{(u_i, b_i)\}_{i=1}^m$*, the revenue-maximizing price is* $\hat{\rho} = \bar{u}_{i^*}^{-1}$ *for some* $i^* \in [m]$.

*Proof.* It suffices to show that the set of all local maxima of revenue (as a function of $\rho = u^{-1}$) correspond exactly to the set of points $\{\bar{u}_i^{-1}\}_{i=1}^m$. Assume w.l.o.g. that $\bar{u}_i^{-1}$ are ordered. Then for any interval $(\bar{u}_i^{-1}, \bar{u}_{i+1}^{-1})$, revenue is linear in $\rho$; this is since for all $\rho$ in this interval, the set of users that purchase are precisely $j = 1, \ldots, i$, each purchasing $u_j$ units at price $\rho$. Next, notice that at any $\rho = \bar{u}_i^{-1}$, increasing $\rho$ infinitesimally causes $i$ to *not* purchase, since she can no longer afford her required units $u_i$ at budget $b_i$, and revenue exhibits a sharp drop. Thus, revenue is discontinuous peicewise linear with increasing segments between the $u_i^{-1}$. $\square$

Fig. 2 illustrates the structure of revenue as a function of price. Thm. 1 implies that it suffices to compute $r$ at prices $\rho_i = 1/\bar{u}_i$ for all $i \in [m]$, choose the maximizing $i^*$, and set $\hat{\rho} = \rho_{i^*}$. We will henceforth refer to the user $i^*$ as the *price setter*. Sorting by $\bar{u}_i$ makes this process efficient: at price $p_i$, the set of point who purchase are precisely $j$ for which $\bar{u}_j \leq \bar{u}_i$. Since revenue at $i$ is $p_i U = p_i \sum_{\bar{u}_j \leq \bar{u}_i} \bar{u}_j$, we can update $U$ on the fly as a cumulative count of total units bought, and multiply by price, giving runtime of $O(m \log m)$.

One interesting observation is that prices are agnostic to scale: if we multiply demand (or budgets) for all users by a constant, then prices will scale inversely (proportionally). Such "change of currency" has no effect on user responses.

## 4. Learning Approach

We now turn to the question of how to learn an accurate classifier on the market distribution it induces. Our general approach will be to follow the empirical risk minimization framework and replace the expected risk in Eq. (7) with an

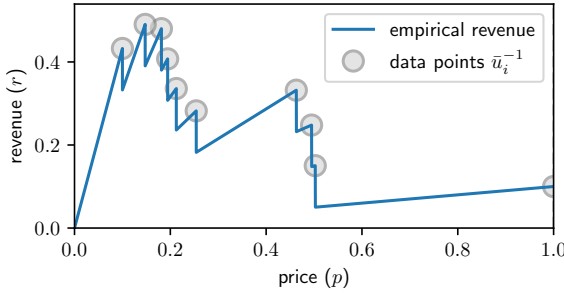

Figure 2: **Empirical revenue as a function of price.** Revenue increases before each $\bar{u}_i^{-1}$ and drops immediately after, implying the argmax is attained at some $i^* \in [m]$ (Thm. 1).

empirical proxy objective over the sample set $S$, namely:

$$\underset{h \in H}{\operatorname{argmin}} \frac{1}{m} \sum_{i=1}^{m} \ell(x_i^h, y_i; h) + \lambda R(g), \quad x_i^h = \Delta_h^{\mathtt{market}}(x_i, b_i)$$
(13)

Here $\ell$ is a surrogate loss (e.g., hinge), $R$ is an (optional) regularization term with coefficient $\lambda$, and responses $\Delta_h^{\mathtt{market}}$ are defined w.r.t. empirical market prices, $\hat{p} = p^*(h; S)$.

**Challenges.** There are several challenges to optimizing Eq. (13). First, as in standard strategic classification, $\Delta$ is an argmax operator, which is non-differentiable and even discontinuous. Second, in our market setting, the objective no longer decomposes over examples, since how each $x_i$ moves now depends on *all* points in $S$ through $\hat{p}$. Third, prices $\hat{p}$ depend not only on the data, but are also a function of $h$, which is the target of optimization. Finally, it is unclear what are appropriate choices for $\ell$ and $R$ since strategic learning often requires using specialized proxy losses and regularizers.

**Approach.** Our solution will be to replace $\Delta_h^{\mathtt{market}}$ with a differentiable proxy that permits to take gradients 'through the market'. First, notice that *conditioned on prices*, user updates $x_i^h$ become independent; this is precisely role prices play in any efficient market. Next, we define $\ell$. While there are no known strategic losses for linear costs—even with fixed parameters—surprisingly we show it is possible to adapt the *strategic hinge* (Levanon & Rosenfeld, 2022):

$$\ell_{\text{s-hinge}}(x, y; h) = \max\{0, 1 - y(w^\top x + \tau + 2\|w\|)\} \quad (14)$$

which applies to fixed 2-norm costs. Although our costs are linear, since we work with market prices of the form $p = \rho w$, users are modelled as moving towards the decision boundary. Thus, because prices adapt to the classifier $h$, users respond to (directional) market prices $p^h$ 'as if' projected onto $h$, in the same manner as under (symmetric) 2-norm costs. Note $\ell_{\text{s-hinge}}$ penalizes all points according to the maximal moving distance of 2 (for $y \in \{\pm 1\}$), which is the same for all users. The key difference in our setting is that users have individualized maximal distances: for a

given $\rho$, this is the amount of units that each user $i$ can buy, namely $b_i/\rho$. This gives our proposed *market hinge loss*:

$$\ell_{\text{m-hinge}}(x, y; h, \rho) = \max\{0, 1 - y(w^\top x + \tau + \frac{b}{\rho}\|w\|)\}$$
(15)

A primary benefit of $\ell_{\text{m-hinge}}$ is that it does not include $\Delta$ explicitly. Furthermore, it requires only the scalar market price $\rho$ (which depends on $h$). Our final step is to replace $\rho$ with a *differentiable market price* $\tilde{\rho}$ as a smooth approximation. This is achieved by making Algo. 1 itself differentiable—for full details see Appendix. C. The relation between the market hinge and the 0-1 loss is explored in Appendix B.1.

## 5. Learning in Markets: Exploratory Insights

In this section we use synthetic experiments to demonstrate the basic mechanics underlying how learning creates markets, and how induced markets affect learning outcomes. As we show, such effects can be quite stark. We begin with questions regarding fixed $h$, and then consider $h$ that are learned.

### 5.1. Typical market behavior

Given a classifier $h$, how will the market respond? Let $q$ be the distribution over demand-budget pairs $(u, b)$ induced by $h$. By Sec. 3, $q$ fully determines prices. For our following analysis we will focus on $q$ that are simple, parametric, and well-behaved. We first consider uniform budgets, and then move to heterogeneous budgets that vary across users.

**Uniform budgets.** When $b = 1$ for all users, and so $\bar{u} = u$, users are naturally 'ordered' by their demand. From Sec. 3, this means that if $u^*$ is the revenue-maximizing point (i.e., the price setter), then all users with $u \leq u^*$ move, and all others do not. This is similar to the standard setting (with fixed costs and uniform budget), only that the threshold on who will move is now adaptive ($u^*$) rather than fixed ($\tau$).

Since only the closest users on the negative side of $h$ move, the main question here is *how many* of them will be deemed as sufficiently close by the market. Fig. 3 shows revenue curves for several demand distributions, depicting the subset of negatively-classified points, $p(x \mid \hat{y} = 1)$. Here we use various parameterizations of the expressive Beta family, scaled to $[1, 10]$. The figure also shows for each distribution the price setter $u^*$ and the percentage of points that cross (i.e., all $u \leq u^*$). Although the distributions are quite diverse in shape, market prices are typically low, and price-setters lie mostly in extreme upper quantiles. As a result, *almost all points cross*, with over $95\%$ in most cases. This effect is robust across many distributions—see Appx. A.2.

To gain some intuition as to the underlying reason, the following result provides a simple sufficient condition:

**Theorem 2.** *Let $q_f(u)$ be a demand distribution with pdf*

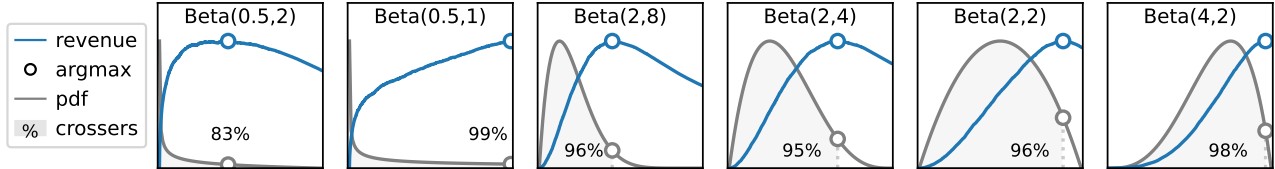

Figure 3: **Demand and price setters.** Demand distributions $q(u)$ for various Beta distributions and $b = 1$. Shown are pdfs and revenue curves. Note how revenue-maximizing points ('price setters') are extreme, suggesting that almost all points cross.

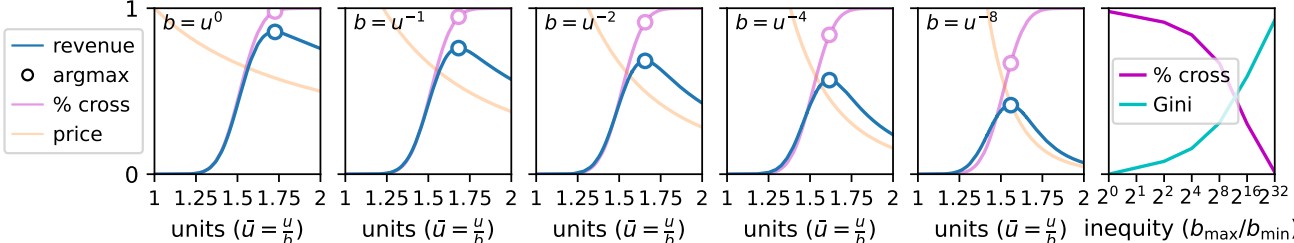

Figure 4: **Demand under varying budgets.** When budgets $b$ decrease as demand for units $u$ grows, price setting points become less extreme. However, this effect is mild, and only very high inequity (Gini$\approx$1) helps to suppress mass crossing.

$f(u)$. *Then if the function $f(u)u$ is either strictly increasing, decreasing, or unimodal, it holds that:*

1. *There is a unique revenue-maximizer $u^*$.*

2. *Let $u_{\max} = \mathrm{argmax}_u f(u)u$, then $u^* \geq u_{\max}$.*

Since $f(u)u$ is unimodal under any log-concave $f$, Thm. 2 applies to many known distribution classes.[5] Appendix A.3 includes a proof and an in depth analysis of some examples.

**Correlated budgets.** When users vary in budgets (and so $u \neq \bar{u}$), this can be thought of as 'distorting' demand by scaling units as $u \mapsto \frac{1}{b}u$. Note this means that far-away points can now be closer, and close points can move far away, depending on $b$. Potentially, this can lead to less extreme price setters if the distribution becomes concentrated around smaller values; this effect occurs mildly for $\mathrm{Beta}(0.5, 2)$ in Fig. 3, which is left-skewed. Because demand is now over $\bar{u} = u/b$, then if we think of $b$ as a function of $u$, demand will be skewed if $b$ is sub-linear in $u$, since this will "push" larger $\bar{u}$ increasingly further. If this negative correlation is sufficiently strong, then market prices should be higher, and we can expect fewer points to cross. Fig. 4 shows revenue, prices, prices-setters, and the percentage of crossers for $b = u^{-\alpha}$ with $\alpha \in \{0, 1, 2, \ldots, 32\}$. Here we use Gaussian $u$ scaled to $[1, 2]$, so that $b_{\min} = 1$ for the smallest $u$, and $b_{\max} = 2^{-\alpha}$ for the largest. Results show that increasing $\alpha$ does shift the price setter, and reduces the number of crossers. However, this requires $\alpha$ to be large, and even for $\alpha = 16$, 30% of points still move.

[5]This includes: Normal, uniform, exponential, logistic, Laplace, Gamma, Beta, Weibull, Gumbel, Rayleigh, and Chi$^2$ distributions.

**Implications.** If $h$ is such that most points are able to move, then this can have dire implications on predictive performance. Because learning generally aims to separate points by their class $y$, for any moderately accurate classifier the majority of points that will participate in the market (i.e., have $\hat{y} = 0$, and therefore $u > 0$) will be negative ($y = 0$). This means that for classifiers with high *pre*-market accuracy, we can expect performance to drop to as low as $\sim$50% *after* the market forms. This ill effect can be somewhat mitigated if budgets correlate with distance to $h$, but *only if inequity is extremely high* within negative points. Fig. 4 (right) shows the relation between the ratio of crossers and inequity in budgets, measured in Gini units, as a function of $\alpha$; in our example, high accuracy is possible only when the gap between the lowest and highest budgets is an extreme $2^{32}$-fold.

### 5.2. Market-aware thresholds

Strategic movement by negative points is harmful to accuracy; but positive points that move are actually *helpful* since they correct the classifier's mistakes. In standard strategic classification, a useful strategy that exploits this idea is to 'raise the bar' by increasing $\tau$ for a given $h = h_{w,\tau}$. Unfortunately, this idea does not easily transfer to a market setting, because *prices adapt to changes in $\tau$*. Nonetheless, varying $\tau$ for a given $h$ can still have a positive effect.

Our next construction allows to accommodate for changes in $\tau$. Let $h$, and w.l.o.g. assume $h = h_{w,0}$. Define $p(z)$ as the induced distribution of distances $z$ from points $x$ to the decision boundary of $h$. For any $\tau$, we can express the marginal over units as $q_\tau(u) = p(z \mid z \leq \tau)$. We can now ask how $\tau$ shapes demand. Our next example sets $p(z)$ to be

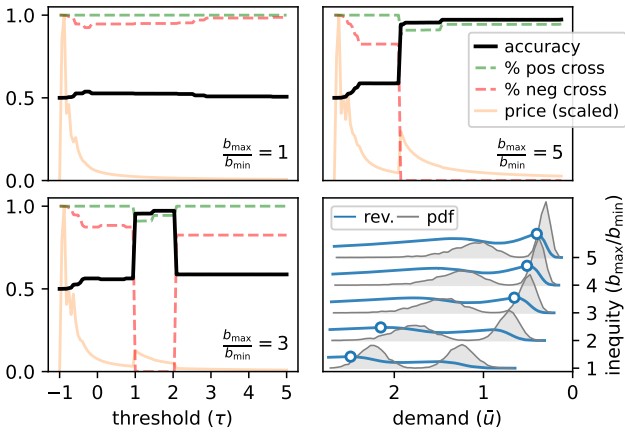

Figure 5: **Varying threshold.** For a mixture distribution of two class-conditional Gaussians, $p(x\,|\,y) = \mathcal{N}(y\mu, \sigma)$, varying the threshold $\tau$ results in surprising outcomes under induced market responses. For uniform budgets (top left), there is no good solution. When inequity in budgets $b$ is moderate (top right), accuracy jumps to 1 once a critical point is reached. When it is low (bottom left), this occurs only at a small interval. Increased inequity distorts the demand distribution, at some point enabling accuracy $\approx 1$ (bottom right).

a mixture of two class-conditional Gaussians $p(z\,|\,y)$, where $p(z\,|\,y=0)$ is scaled to $[-1,0)$ and $p(z\,|\,y=1)$ is scaled to $(0,1]$. In terms of accuracy, we would like $\tau$ to cause points from $p(z\,|\,y=1)$ to move, and from $p(z\,|\,y=0)$ to stay unchanged. Fig. 5 shows prices, accuracy, and the ratio of crossers per class for the range $\tau \in [-1, 5]$. When budgets are uniform ($b=1$), no threshold obtains accuracy above 55% *despite the data being separable*. This is because increasing $\tau$ causes prices to decrease and remain low enough so that almost all points cross; essentially the same effect of Sec. 5.1. However, when $b$ negatively correlates with $z$—even mildly—then it becomes possible to achieve high accuracy: for $b$ that increases linearly with $z$ from $b_{\min} = 1$ to $b_{\max} = 5$, we see that once $\tau \approx 0.75$, accuracy abruptly jumps from $\sim 0.5$ to $\sim 0.95$. This is because at this threshold, the price setter shifts from being an extreme point of $p(z\,|\,y=-1)$ to an extreme point of $p(z\,|\,y=1)$ (see ridgeline plot). Note that this holds for *any* $\tau \geq 0.75$; here the adaptivity of prices plays in favor of learning and provides robustness to the choice of $\tau$.[6] Interestingly, for a smaller $b_{\max} = 3$, we get that accuracy is high only for $\tau \in [1, 2]$; once $\tau$ becomes too large, the price returns to be set by an extreme negative point.

Fig. 5 (bottom right) reveals an interesting phenomenon: as inequality varies, the price setter jumps from an extreme point of one cluster to that of another. We hypothesize this

---

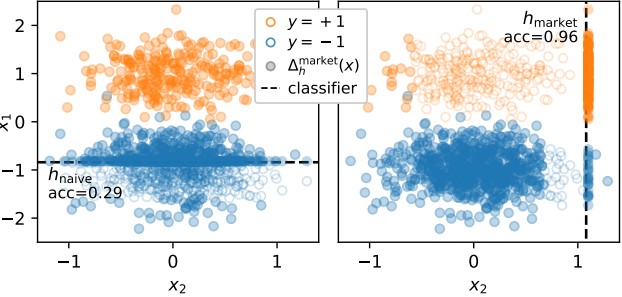

Figure 6: **Market classifiers.** Consider a distribution where $x_1$ enables class separation, and $x_2$ is uninformative of $y$. A naïve classifier that uses only $x_1$ is unable to prevent negative points from crossing, and attains low accuracy (left). In contrast, a market-aware classifier that uses $x_2$ is able to capitalize on the variation in budgets to classify well (right).

clustering behavior applies widely—see Appendix B.3.

### 5.3. Market-aware classifiers

In terms of the market, control over $h$ allows the learner to 'choose' which demand distribution $q$ to work with: the choice of $w$ determines $p(z)$, and $\tau$ induces $q(u)$. This gives learning much power over which users will be in the market, as well as which of them will cross. Interestingly, an $h_{w,\tau}$ that is effective on the induced market need not be accurate on raw data $(x, y) \sim D$. For example, $w$ can focus weight on features that are entirely uninformative of $y$. Our next construction demonstrates an extreme version of this idea.

Let $d = 2$, and consider $(x_1, x_2) \sim D$ composed of per-feature class-conditional Gaussians $D(x_i\,|\,y)$. The first feature is $x_1 \sim \mathcal{N}(\mu y, \sigma)$, which allows to separate the classes (we use $\mu = 1$ and $\sigma = 0.15$). The second feature is $x_2 \sim \mathcal{N}(0, \sigma)$, i.e., has the same distribution under both classes, and so is by definition inseparable. We set $D(y=0) = 0.75$ and $D(y=1) = 0.25$. Here we let $b$ depend on labels as $b = 1 + 4y$. Fig. 6 shows the behavior of two classifiers on this data: $h_{\text{naïve}}$ which uses only $x_1$ (left), and $h_{\text{market}}$ which uses only $x_2$ (right). The idea of $h_{\text{naïve}}$ follows that of Sec. 5.2: separate the raw data well, and then tune $\tau$ on the market. Here we see that this approach breaks down completely: the best it can achieve is 0.29 accuracy, since it cannot prevent the bulk of negatives from crossing. In contrast, $h_{\text{market}}$ exploits the market to *create* separability over the otherwise ineffective $x_2$. This is the optimal market-aware classifier. The correlation between $b$ and $y$ results in a demand distribution $q$ which clusters the positive $\bar{u}$ close to $h$, and pushes negative $\bar{u}$ sufficiently far. This results in almost only positive points crossing, and accuracy reaches 0.96. In Appendix B.4 we show this effect can be even more extreme. Note learning will not always tend to favor $h$ in which labels and budgets correlate—see Appendix B.2.

---

[6]This is in contrast to standard strategic classification which requires a particular $\tau$ and can be highly sensitive to its choice.

# 6. Experiments

We now turn to demonstrate how our market-aware strategic learning framework performs empirically on real data with simulated market behavior. We use two datasets common and publicly available datasets and adapt them to our strategic market setting: (i) the `adult` dataset, showed here, and using `capital_gain` feature as a proxy for budgets $b$; and (ii) the `folktables` dataset, deferred to Appendix B.5. For further details see Appendix. D.1. Code is publicly available at https://github.com/BML-Technion/MASC.

**Setup.** Our method of market-aware strategic classification (`MASC`) optimizes the proxy objective proposed in Eq. (13)—see Appendix D.3 for details. We compare to two baselines: (i) `naïve`, a conventional non-strategic classifier; and (ii) `strat`, a strategic classifier that anticipates user responses (to fixed prices) but does not account for how the market adapts.[7] The latter is done by training `naïve`, computing optimal prices $p$, setting $w = p$, and then optimizing $\tau$ to maximize accuracy on a held-out set. We measure short-term performance on current prices $p$ (`short`) and long-term performance after prices re-equilibrate at $p^h$ (`long`). We also show the accuracy of `naïve` on non-strategic data (for which it is consistent) as a benchmark.

Our main question regards the effect of budget distribution on learning and its outcomes. The distribution of budgets $b$ as found in the data is highly skewed, with a ratio of $b_{\min}$ to $b_{\max}$ of approximately 1:1000, which depicts a state of high inequality. To balance this, we consider the effects of redistributing budgets to attain *lower* inequality, achieved by rescaling budgets to reduced ranges $[1, 2^\alpha]$ for $\alpha = 2, \ldots, 10$, where the largest value of $\alpha_{\text{true}} = 2^{10} \approx 1000$ matches the original data (star marker). For each such $k$, we measure test accuracy, welfare (normalized by total budget), and social burden (Milli et al., 2019) (normalized by total budget of positives). We also measure the ratio of positive points that move (high is good) and of negative points that don't (low is good). All results are averaged over 10 random splits, and we report mean and standard errors.

**Results.** Fig. 7 shows results across budget redistributions at different inequity scales $\frac{b_{\max}}{b_{\min}} = 2^2, \ldots, 2^{10}$. In terms of accuracy (left), all methods improve as budget gaps increase, but at different rates. `MASC` clearly outperforms `naïve` by a large margin. It also outperforms `strat` in the long term for almost all scales $\alpha > 2^4$ (which includes $\alpha_{\text{true}}$) and shows similar performance for the lowest scales. Interestingly, this is where `strat` reveals a large gap between short-term performance (for which it is optimized) and long-term outcomes (as a result of prices adapting to updated demand).

Fig. 7 (top right) depicts the percentage of users that cross per class. Note the abrupt drop in negative crosses at $\alpha \approx 2^4$,

---

[7]This draws on the idea of the main algorithm in Hardt et al. (2016).

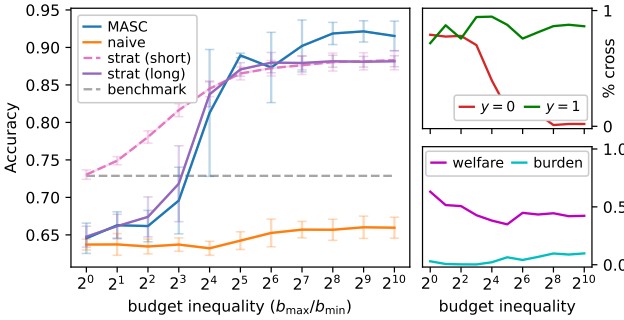

Figure 7: **Results on `adult`. (Left:)** Accuracy across reduced budget inequity scales, relative to the data's original highly skewed scale of $\frac{b_{\max}}{b_{\min}} \approx 2^{10}$ (star). **(Right:)** For `MASC`, per-class ratio of crossers (top; high is good for $y = 1$, low is good for $y = 0$), welfare, and social burden (bottom).

which matches the sharp performance increase for `MASC`. This shows that larger scales make it possible to find a classifier for which negative points are mostly unable to cross. A possible explanation is our clustering hypothesis: once accuracy is sufficiently high, the price setter jumps from an extreme negative point (under which almost all points move) to an extreme positive point (under which mostly positives move). In terms of social outcomes, welfare (normalized) begins reasonably high, but reduces to $\sim 0.5$, and does not fully recover. Burden remains flat until scale $2^4$, and then gradually rises, but remains relatively low throughout.

# 7. Discussion

The use of learned models to inform decisions about humans has become common practice. But when those very humans also take interest in prediction outcomes, conventional learning tools no longer necessarily apply. This paper advances the idea that when users seek to obtain certain predictions, learning inevitably becomes a driver of demand. When this creates an opportunity for profit, it is only natural to expect that a market will form. Learning classifiers that induce markets poses unique challenges as a learning task. Our paper takes a first step to address these, and so targets a particular market setting and pursues a basic understanding of it. But there is of course a plethora of other market settings to explore at this new intersection of machine learning and markets. Nonetheless, the idea that learning can drive economic outcomes has broader implications to consider. One example is the question of how learning influences social welfare, as it relates e.g. to market efficiency. Another example is the question of information asymmetry and the capacity of learning to exploit its informational advantage. Given the growing influence of learning on our lives, such questions merit careful thinking and much deliberation. Our hope is therefore not only to spark interest, but to also motivate discussion on these important and timely topics.

## Acknowledgements

The authors would like to thank Inbal Talgam-Cohen and Moran Koren for their diligent advice and insightful feedback. This work is supported by the Israel Science Foundation grant no. 278/22.

## Impact Statement

Our paper sets out to study the interplay between learning classifiers and the markets this process can facilitate. We believe that the impact of prediction on economic outcomes can be significant and widespread when machine learning tools are used in social contexts. In the market model we propose, the choice of classifier is modeled as affecting both users and sellers: it inadvertently determines who must invest to be classified as positive (i.e., receive the loan or get the job), what this will cost, and which sellers will profit. These forces arise naturally through how the market coordinates supply and demand. But whereas the mechanics of conventional markets are well understood both in theory and practice, we believe that the role of learning in markets, and the impact that learning can have, has so far been insufficiently explored.

An understanding of how learning creates and affects markets can be used to advance efficient and fair trade, foster equal opportunity, and promote social welfare. It can also be used to gain insight as to how learning-driven markets should be regulated and by what means. But such knowledge and tools should be used with care, as they can potentially serve to drive markets to undesired outcomes. One example is economic inequity, which can be exacerbated by learning, as our results suggest can happen. Another example is information asymmetry: Our stylized market model assumes perfect information and efficient prices. But in a reality where learned models have access to an unparalleled amount of data—certainly more than is accessible to users or sellers—the learning system gains a distinct informational advantage. It is widely recognized that such settings can lead to the exploitation of consumers and even to market collapse (e.g., Akerlof, 1978). We hope that our work serves to encourage fruitful discussion on these important topics.

It is also important to note that the market model we study is simple and draws on many assumptions, such as unlimited supply, a fixed number of exclusive sellers, and no externalities. Results regarding market outcomes, both theoretical and empirical, should therefore be taken under this light. At the same time, we hope this motivates researchers in both machine learning and economics to deepen our understanding of learning and markets in broader and more realistic economic settings.

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

# A. Market prices – additional theoretical and empirical results

## A.1. Equilibrium prices

Consider some classifier $h = h_{w,\tau}$, and assume there exists some $i \in [d]$ for which $w_i > 0$. W.l.o.g. let $i = 1$. The following result states that computing market outcomes under $h$ can be done by (i) projecting demand onto the direction of $w$ (i.e., by computing distances from negatively classified points to the decision boundary of $h$), (ii) computing a (scalar) revenue-maximizing price $\rho^\perp$, and (iii) moving points iff $b > \rho^\perp$. This suggests that even for $w$ with some negative entries, market outcomes can be computed "as if" users move directly towards $h$, i.e., as determined by "prices" $\boldsymbol{p}^\perp = \rho^\perp \cdot w$.

The idea is to show that outcomes are mostly invariant to the actual direction in which points move. Because features are substitutable, we can artificially constrain purchases to a single feature (here, $x_1$) and maintain the same outcomes. This results from the same user taking on the role of prices setter irrespective of the chosen direction of movement.

**Proposition 2.** *Given $h$, let $\rho^1$ by the revenue-maximizing price assuming there is demand only for feature $x_1$, i.e., points can only buy $x_1$ and therefore only move along this dimension. Define $\boldsymbol{p}^1 = (\rho^1, 0, \ldots, 0)$. Then:*

1. *Total revenue will be the same under $\boldsymbol{p}^\perp$ and $\boldsymbol{p}^1$.*

2. *The same set of users will cross $h$ under $\boldsymbol{p}^\perp$ and $\boldsymbol{p}^1$.*

*Proof.* The $\ell_2$ distance of a point $x$ to the hyperplane defiend by $h$ is given by:

$$u = \left| \frac{w^\top x + \tau}{\|w\|} \right|$$

Contrarily, the distance of a point to a hyperplane in the direction of $x_1$ alone is:

$$u^1 = \left| \frac{w^\top x + \tau}{w_1} \right|$$

which we think of $u^1$ as units of demand in the direction of $x_1$. Together, we get the relation:

$$u = u^1 \frac{|w_1|}{\|w\|}$$

Plugging the above into the definition of maximal revenue gives:

$$
\begin{aligned}
r &= \operatorname*{argmax}_i \frac{b_i}{u_i} \sum_{j: \frac{b_j}{u_j} \geq \frac{b_i}{u_i}} u_j \\
&= \operatorname*{argmax}_i \frac{b_i}{u_i^1 \frac{|w_1|}{\|w\|}} \sum_{j: \frac{b_j}{u_j^1 \frac{|w_1|}{\|w\|}} \geq \frac{b_i}{u_i^1 \frac{|w_1|}{\|w\|}}} u_j^1 \frac{|w_1|}{\|w\|} \\
&= \operatorname*{argmax}_i \frac{b_i}{u_i^1} \sum_{j: \frac{b_j}{u_j^1} \geq \frac{b_i}{u_i^1}} u_j^1 \\
&= r^1
\end{aligned}
$$

where $r^1$ is the total revenue when only purchases of $x_1$ are permitted. The equality holds since scaling does not change the argmax, and by multiplying both sides of the inequalities in the summation by $\frac{|w_1|}{\|w\|}$. Thus, total revenue remains the same.

Note also that by switching from $u$ to $u^1$, albeit scaling, the summation is taken over the same set of points. In other words, a different direction might entail a different currency, but the market will maintain its operation. Thus, the set of points that move also remains the same.

□

For a given $w$, one implication is that for any direction $i$ in which $w_i > 0$, the corresponding $\boldsymbol{p}^i$ is an equilibrium price. When multiple such $i$ exist, any $\boldsymbol{p}_i$ is an equilibrium price. When $w > 0$ in all entries, $\boldsymbol{p}^*$ is also an equilibrium price. Our result above states that market outcomes are equivalent under any of these prices, and so in principle we are free to work with any of them. The second convenient implication is that even for $w$ having negative entries, and for which $\boldsymbol{p}^*$ is not well-defined, we can nonetheless work with $\boldsymbol{p}^\perp$.

A minor comment is that although we assume only that items and prices are positive (and not $w$), it is possible to constrain $w > 0$ as part of the learning algorithm. Another possibility is to permit negative prices: these can be interpreted as paying to reduce $x$ (e.g., pay the gym to lose weight), but requires constraining $x^h \geq 0$ (or permitting negative $x$). For $w_i = 0$, we can interpret this as seller $i$ not permitted (or able) to sell at all.

### A.2. Expected prices

In Sec. 5.1 we have empirically shown that for many 'natural' distributions over demand, there is: (i) a unique revenue-maximizing point $u^*$, and (ii) this point tends to materialize at extreme quantiles of the distribution. Here we show that this phenomenon holds more broadly. Fig. 8 shows pdf-s, revenue curves, and price setters for a wide range of parameterizations of the $\mathrm{Beta}$ distribution. These include symmetric, left-skewed, right-skewed, concave, bell-shaped, and uniform distributions. For all distributions considered, the price setter is at least in the 80-th percentile, and typically much more extreme.

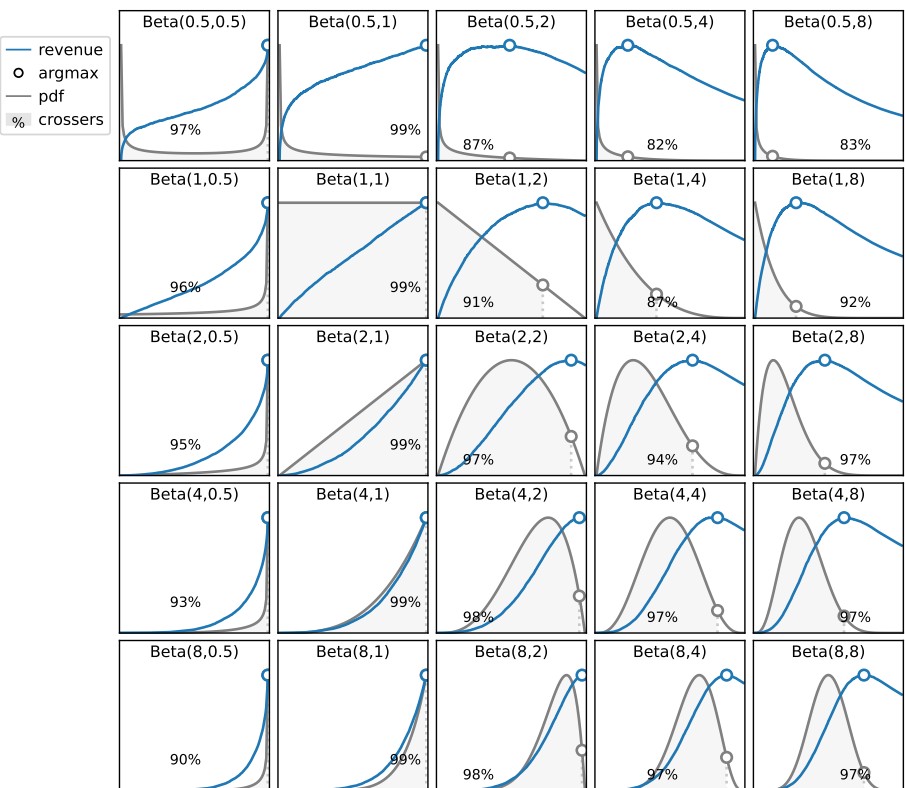

Figure 8: Price setters are extreme across a wide range of $\mathrm{Beta}$ distributions.

### A.3. Theoretical insight

To complement the observations above, this section aims to provide a theoretical underpinning for the questions of (i) when do unique revenue-maximizing prices exists, (ii) why is this prevalent across many natural distributions, and (iii) how extreme are price setters (e.g., in terms of quantiles). We begin with some general claims and sufficient conditions, and then present some examples of particular distribution classes which we analyze in depth.

### A.3.1. ANALYZING REVENUE

Consider a continuous distribution over (univariate) revenue defined by a pdf $f(u)$. We assume that $f$ has support on $[0, t]$ for $t \in \mathbb{R}_+ \cup \{\infty\}$, and consider uniform budgets $b = 1$ for all users. This is made w.l.o.g.—see below. Recall that expected revenue $r(u; f)$ is defined as the sum of demands of all users $u' \leq u$, divided by $u$. This can be rewritten as:

$$r(u; f) = \frac{1}{u} \mathbb{E}_u[u'|u' \leq u] = \frac{1}{u} \int_0^t f(u') \cdot \mathbb{1}\{u' \leq u\} \cdot u' du' = \frac{1}{u} \int_0^u f(u') \cdot u' du'$$

To determine whether the expected revenue has a maximum, we compute the derivative of $r(u; f)$ with respect to $u$:

$$r'(u; f) = \frac{d}{du} r(u; f) = \frac{1}{u} f(u) \cdot u - \frac{1}{u^2} \int_0^u f(u') \cdot u' \, du' = f(u) - \frac{1}{u^2} \int_0^u f(u') \cdot u' \, du'$$

Denote by $u^*$ the revenue maximizer w.r.t. $f$ (if such exists). That is, $u^* = argmax_u r(u; f)$. The following observations will be useful:

**Observation 2.** *If $r'(u; f) > 0$ through $0 \leq u \leq t$, then $u^*$ is unique and is attained at $t$.*

**Observation 3.** *If $r'(u; f) < 0$ through $0 \leq u \leq t$, then $u^*$ is unique and is attained at $0$.*

Setting $r'(u; f) = 0$, we find:

$$f(u) = \frac{1}{u^2} \int_0^u f(u') \cdot u' du'$$

$$f(u)u^2 = \int_0^u f(u') \cdot u' du'$$

### A.3.2. PROOF OF THEOREM 2

Theorem 2 states that sufficient conditions for the existent of a unique argmax for expected revenue are that $f(u)u$ is either strictly increasing, strictly decreasing, or strictly unimodal. We now turn to its proof.

*Proof.* Denote by $D(u)$ the function $D(u) = f(u)u$. We split the proof to two distinct cases:

**Case I:** $D(u)$ **contains one maxima point.** Let $\hat{u}$ denote the maxima point of $D(u)$, meaning that $\hat{u} = \text{argmax}_u D(u)$. For $u \in [0, \hat{u}]$, $D(u)$ is increasing, and for $u \in [\hat{u}, t]$, $D(u)$ is decreasing. Therefore, for $0 < u_1 < u_2 < \hat{u}$, it holds that $D(u_1) = f(u_1)u_1 < f(u_2)u_2 = D(u_2)$. Thus, for all $u < \hat{u}$:

$$D(u)u = f(u)u^2 > \int_0^u f(u')u', du'$$

This implies:

$$r'(u; f) = f(u) - \frac{1}{u^2} \int_0^u f(u')u', du' > 0$$

If $f(u)$ is continuous on $[0, t]$, then $r'(u; f)$ is also continuous on $[0, t]$. Therefore, if no $u$ satisfies $f(u)u^2 = \int_0^u f(u')u' \, du'$, then $r'(u; f) > 0$ throughout $[0, t]$. By observation 2, $u^* = t$ maximizes the revenue of the distribution.

Suppose there exists a point $u^*$ such that:

$$f(u^*)u^{*2} = \int_0^{u^*} f(u') \cdot u' \, du'.$$

First, it follows directly that $u^* \geq \hat{u}$, since for $u < \hat{u}$, it holds that $r'(u; f) > 0$. Second, referring to Figure 9, for each $\epsilon > 0$, moving to $u^* + \epsilon$ increases area 1 and decreases area 3. Area 2 changes as well but is mutual to both terms. Therefore, in the range $(u^*, t]$, the following inequality holds:

$$f(u)u^2 < \int_0^u f(u') \cdot u' \, du'$$

Figure 9: Areas 1 and 2 contribute to $\int_0^{u^*} f(u')u'\,du'$, while areas 2 and 3 yield $f(u^*)u^{*2}$. At $u = u^*$, we have $\int_0^{u^*} f(u')u'\,du' = f(u^*)u^{*2}$, which implies that area 1 equals area 3. Furthermore, for each $\epsilon > 0$, shifting to $u^* + \epsilon$ increases area 1 and decreases area 3. Area 2 also changes but remains common to both terms. Thus, for each $\epsilon > 0$, at $u_\epsilon = u^* + \epsilon$, it follows that $\int_0^{u_\epsilon} f(u')u'\,du' > f(u_\epsilon)u_\epsilon^2$.

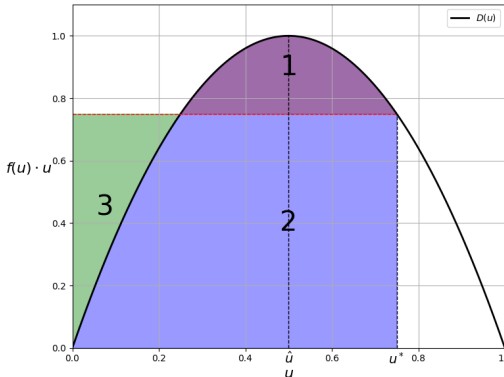

This implies that within the range $(u^*, t]$, the derivative $r'(u; f) < 0$. Consequently, $u^*$ is the unique revenue maximizer by definition.

**Case II:** $D(u)$ **contains no maximum point.** If $D(u)$ is strictly increasing over $[0, t]$, it follows that $f(u)u^2 > \int_0^u f(u')u'\,du'$ for all $u \in [0, t]$. Consequently, $r'(u; f) > 0$ for all $u$, which, by Observation 2, indicates that the unique revenue maximizer is $u^* = t$. Moreover, in this case $\operatorname{argmax}_u D(u) = t$, meaning that $u^* = \operatorname{argmax}_u D(u)$

Conversely, if $D(u)$ is strictly decreasing over $[0, t]$, it follows that $f(u)u^2 < \int_0^u f(u')u'\,du'$ for all $u \in [0, t]$. Consequently, $r'(u; f) < 0$ for all $u$, which, by Observation 3, indicates that the unique revenue maximizer is $u^* = 0$. Moreover, in this case $\operatorname{argmax}_u D(u) = 0$, meaning that $u^* = \operatorname{argmax}_u D(u)$. □

We note that the proof can be easily extended to distribution in the range $[a, t]$, for $a > 0$. The changes to the original proof are minor, and include modifying the lower limit of the integration. For distributions in range $[a, \infty]$, the proof is valid too, as long as $D(u)$ is strictly increasing, decreasing, or unimodal.

The proof also give a lower bound on $u^*$ in terms of $f$:

**Corollary 1.** *Under all conditions of Thm. 2, it holds that $u^* \geq \hat{u}$.*

The following examples show this relation explicitly for two classes of distributions: Beta, and uniform.

### A.3.3. EXAMPLE: BETA DISTRIBUTION

Based on Theorem 2, we establish the following result about Beta distributions:

**Theorem 3.** *For every $a, b > 0$, let $f(u)$ denote the probability density function (PDF) of the Beta distribution Beta$(a, b)$, defined on the interval $[0, 1]$. Then, the function $D(u) = f(u) \cdot u$ is either strictly increasing or strictly unimodal.*

The following theorem implies that every Beta distribution has a unique revenue maximizer, which is confirmed empirically over different Beta distributions in Figure 8.

*Proof.* The PDF of Beta$(a, b)$ is given by:

$$f(u) = K_{a,b} u^{a-1}(1 - u)^{b-1},$$

where $K_{a,b} > 0$ is a normalization constant that depends only on $a$ and $b$. Thus, $D(u) = f(u) \cdot u = K_{a,b} u^a (1 - u)^{b-1}$. We will show that $D(u)$ is either strictly increasing, or strictly unimodal. First, compute the derivative of $D(u)$ w.r.t. $u$:

$$\frac{d}{du} D(u) = K_{a,b} \left[ a u^{a-1}(1 - u)^{b-1} - u^a (b - 1)(1 - u)^{b-2} \right]$$

Simplifying the expression gives:

$$\frac{d}{du} D(u) = K_{a,b} u^{a-1}(1 - u)^{b-2} \left[ a(1 - u) - u(b - 1) \right]$$

and setting $\frac{d}{du}D(u) = 0$, we solve:

$$K_{a,b}u^{a-1}(1-u)^{b-2}\left[a(1-u) - u(b-1)\right] = 0$$

The solutions are:

$$u_1 = 0, \quad u_2 = 1, \quad u_3 = \frac{a}{a+b-1}$$

Here, $u_1$ exists if $a > 1$, and $u_2$ exists if $b > 2$ (otherwise, they are undefined due to the powers of $u^{a-1}$ and $(1-u)^{b-2}$). Since the Beta distribution is defined on $[0, 1]$ and $D(u_1) = D(u_2) = 0$, we focus on $u_3$. For $u_3 \in [0, 1]$, it must hold that $b > 1$.

Next, observe that for all $u < u_3$, the term $a(1-u) - u(b-1) > 0$. Therefore, $D'(u) > 0$ and $D(u)$ increases in this range. The proof now splits into two cases:

**Case 1:** $u_3 \notin [0, 1]$. In this case, $D(u)$ is strictly increasing over the interval $[0, 1]$, because $D'(u) > 0$ throughout $[0, 1]$ and the proof is complete.

**Case 2:** $u_3 \in [0, 1]$. For $u_3 < u < 1$, the term $a(1-u) - u(b-1) < 0$. Therefore, $D'(u) < 0$ and $D(u)$ decreases in this range. Thus, $u_3$ is the sole maximum point of $D(u)$, which implies that $D(u)$ is strictly unimodal, as required. ☐

As shown in the proof of Theorem 2, if $D(u)$ is strictly increasing, the revenue maximizer occurs at the right edge of the distribution, which in this case is at $u = 1$. If $D(u)$ is strictly unimodal, we know that the revenue maximizer is greater than the maximum point of $D(u)$. In this case, the maximum point is $\frac{a}{a+b-1}$ (noting that $b > 1$ in this scenario).

Moreover, the maximum point of a Beta distribution with parameters $a, b$ is given by $\operatorname{argmax}_u f(u) = \frac{a-1}{a+b-2}$. For $b > 1$, we obtain:

$$\frac{a}{a+b-1} > \frac{a-1}{a+b-2},$$

which implies that the percentile of $\arg\max_u D(u)$ is greater than the percentile of $\arg\max_u f(u)$, and both are smaller than the percentile of $u^*$.

This analysis provides an intuition for the unequivocal empirical results shown in Figure 8, which demonstrate that the revenue maximizer is at least in the 80th percentile (and typically even higher). To conclude, under this family of distributions, when they induce individual demands for a feature under a uniform budget, a large percentage of users will be able to afford purchasing the amount of the feature they need.

### A.3.4. EXAMPLE: UNIFORM DISTRIBUTION

We now perform a similar analysis for the uniform distribution over the range $[0, t]$, where $t > 0$. The PDF of this distribution is constant for all $u$: $f(u) = \frac{1}{t}$. Consequently, $D(u) = f(u)u = \frac{1}{t}u$ is a strictly increasing function of $u$. By Theorem 2, the revenue maximizer for this distribution is the right edge of the range, which is $t$.

We can extend this result to the uniform distribution over the range $[a, b]$.

**Theorem 4.** *For any $a, b > 0$, let $f(u)$ denote the probability density function (PDF) of the uniform distribution over $[a, b]$. Then, the revenue maximizer is unique and occurs at $b$.*

*Proof.* The PDF of the uniform distribution is constant for all $u$: $f(u) = \frac{1}{b-a}$. The function $r(u; f)$ is therefore given by:

$$r(u; f) = \frac{1}{u}\int_a^u f(u') \cdot u' \, du' = \frac{1}{u}\int_a^u \frac{1}{b-a} \cdot u' \, du' = \frac{1}{u(b-a)}\int_a^u u' \, du'$$

Evaluating the integral:

$$r(u; f) = \frac{1}{u(b-a)}\left[\frac{u^2}{2} - \frac{a^2}{2}\right] = \frac{1}{2(b-a)}u - \frac{a^2}{2(b-a)}\frac{1}{u}$$

Next, compute the derivative of $r(u; f)$ with respect to $u$:

$$r'(u; f) = \frac{1}{2(b-a)} - \frac{a^2}{2(b-a)} \cdot \left(-\frac{1}{u^2}\right) = \frac{1}{2(b-a)} + \frac{a^2}{2(b-a)}\frac{1}{u^2}$$

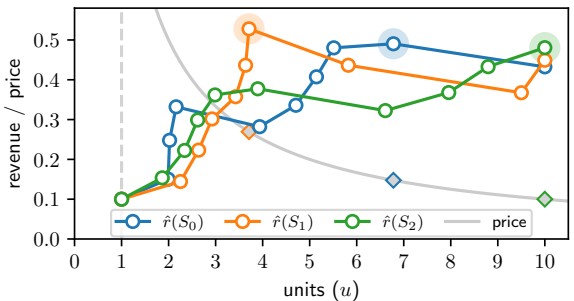
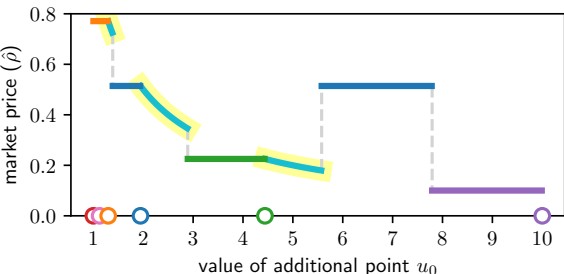

Figure 10: Empirical revenue curves $\hat{r}$ for different samples $S_i$ featuring different revenue-maximizing points $u^*$.

Figure 11: The effect on price of adding a single point to a fixed sample. Original sample points shown on x-axis.

Since both terms are positive for all $u \in [a, b]$, it follows that $r'(u; f) > 0$ for all $u \in [a, b]$

By Observation 2, the revenue maximizer is unique and occurs at the right edge of the range, which is $b$. □

### A.4. Empirical prices

Our analysis above considers expected prices defined over a demand distribution. But in practice, learning must work with finite samples, and therefore with empirical markets. We begin by investigation some features of empirical markets, revenue, and prices, and then make the connection to population markets with expected revenue and prices.

#### A.4.1. THE PRICE-REVENUE LANDSCAPE

Thm. 1 states that the revenue-maximizing price $\hat{\rho}$ is always some $u_i^{-1}$; hence, we can instead think of revenue as a function of inverse prices $\frac{1}{\rho} = u$, measured in demand units. This is useful since we can now consider directly how changes in the demand set affect revenue, and through it, the optimal price.

Fig. 10 plots empirical revenue $\hat{r}(S)$ for three different samples $S_j$ of size $m = 10$ with units $u_i$ scaled to span $[1, 10]$:

Note that revenue always begins at $\frac{1}{m}$ for the smallest $u_i$ since only one unit is sold (to one user) at price $\rho = 1$. From here, however, outcomes can differ considerably across samples, in terms of the shape of the revenue curve, the location and index of the price setter $i^*$, and the optimal price $\hat{\rho}$.

This raises the question: how sensitive are market prices to variation in demand? For this, we take a sample $u_1, \ldots, u_m$, and measure how prices change due to adding a single new point $u_0$. Fig. 11 shows the outcome of this process for a fixed select demand set of size $m = 5$ and for an increasing value of an additional point $u_0 \in [1, 10]$ (x-axis):

The $u_i$ are shown in color and positioned on the x-axis. The revenue curve includes segments colored according to the matching price setter, with turquioise (and yellow highlight) indicating that the price setter is $u_0$. As can be seen, the value of $u_0$ has a stark effect on market prices: even though it is increased gradually, prices jump at discrete points whenever the price-setter $i^*$ changes. Generally, prices are down-trending, and $i^*$ appear in increasing order of $u_i$—but this is not necessary, as prices can also jump up, and some $s_i$ can be price-setters more than once.

#### A.4.2. WHY PRICES JUMP

One reason for this behavior is that optimal prices may not be unique. The following is an extreme construction in which *all* points are revenue-maximizing.

**Proposition 3.** *Let $m \geq 3$, and w.l.o.g. assume uniform budgets. Define $u_1, \ldots, u_m$ recursively as:*

$$u_i = u_2 \cdot \sum_{j < i} u_j, \quad u_2 = 2, \quad u_1 = 1$$

*Then for all $i > 1$, prices $\rho_i = u_i^{-1}$ attain the same revenue.*

Together with Thm. 1, this implies that $\hat{r}$ is also maximized under all $\rho_i$. To see how this lends to price jumps, consider the minimal case of $m = 3$. If we slightly decrease $u_2$, then it becomes the unique price-setter; in contrast, if we slightly increase $u_2$, then $u_3$ becomes the price setter. Thus, small perturbations in $u_2$ can cause prices to jump between $\rho_2$ and $\rho_3$.

### A.4.3. REVENUE FOR LARGE SAMPLES

Our examples above considered mostly very small market sizes. Fortunately, prices tend to be more well-behaved when the number of samples grows as long as the underlying demand distribution is well-behaved. For example, for $u \sim \text{Beta}(0.5, 4)$, (and scaled to $[1, 10]$), Fig. 12 presents revenue curves (left) as well as maximal revenue (top right), and empirical market prices (bottom right) for samples of increasing size $m$:

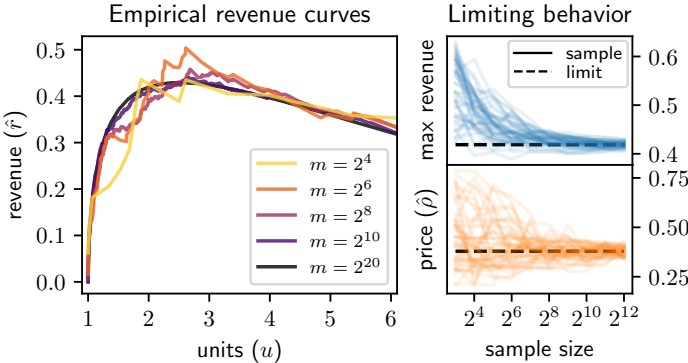

Figure 12: Revenue and prices for increasing sample size.

As can be seen, despite significant variation under small $m$, results stabilize as $m$ grows in terms of the revenue curve (left), its maximum value (top right), and optimal prices (bottom right).

## B. Experiments - additional results

### B.1. Market hinge loss vs. 0-1 loss

Our proposed m-hinge in Sec. 4 permits both tractable optimization via gradient methods and without the need for explicitly computing best-responses $\Delta_h^{\texttt{market}}(x)$. Due to Levanon & Rosenfeld (2022), it also intends to maximize a generalized notion of 'strategic' margin (although their generalization bounds do not immediately carry to our setting due to dependence across users in the sample). Here we examine the loss landscape for the m-hinge under a simple 1D example and compare it to the 0-1 loss, which it intends to approximate.

Data as generated as follows. Setting $d = 1$, we sample each $x$ from a class-conditional distribution $x \sim \mathcal{N}(\mu_y, 1)$. Classes are balanced with $p(y = 0) = p(y = 1) = 0.5$. Budgets $b$ are sample uniformly from $[1, 8.5]$ for negative points and from $[8.5, 16]$ for positive points. This means that $b$ are generally in $[1, 16]$ and correlate with $y$, and so generally with $x$, but do with $x$ given $y$ for either class. We sample 1000 points in each setting.

Fig. 13 illustrates the loss landscape for increasing class mean gaps $\mu_1 - \mu_0 \in \{-1, 0, 1, 2, 3\}$. For a range of thresholds $\tau \in [-3, 10]$, each plot shows values of: (i) the 0-1 loss, (ii) the m-hinge with exact empirical prices $\hat{\rho}$, (iii) the m-hinge with smoothed empirical prices $\tilde{\rho}$ (using a mild $T_{\text{softsort}} = 0.1$ and large $T_{\text{softmax}} = 10$), (iv) prices, and (v) the price setter (in percentage from all points, sorted by their $x$ value). Note prices and m-hinge values are scaled to fit the plot.

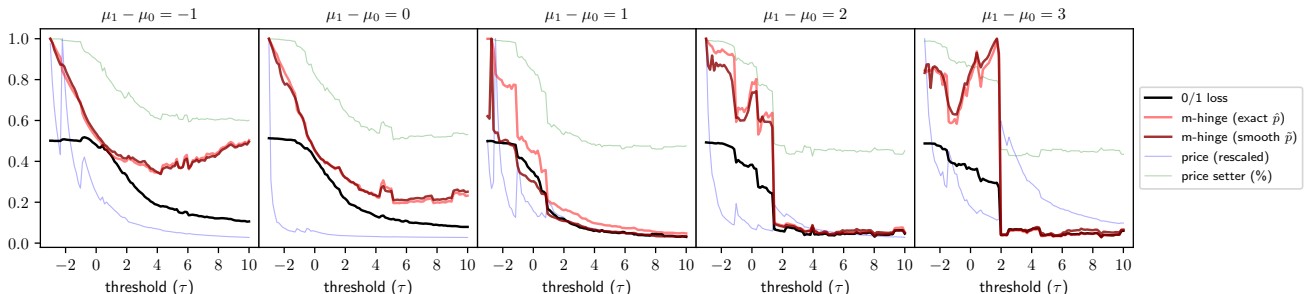

Figure 13: Market hinge loss landscape, compared to the 0-1 loss.

Overall the m-hinge appears to be an adequate proxy for the 0-1 loss. When classes are reasonably distances at $\mu_1 - \mu_0 = 1$ (center), the 0-1 loss decreases gracefully as $\tau$ increases. The m-hinge follows closely, but struggles for negative $\tau$ due to large price variation. Once $\tau$ reaches $\approx 1$, the price setter begins to stabilize at around 60%; from this point on the m-hinge is well-behaved. For larger gaps (right), there is an abrupt jump in the 0-1 loss once a certain $\tau$ is reached. This can be seen both in prices peaking, and the price setter settling on 50%. The 0-1 becomes very close to 0, and the m-hinge is faithful in this regard. Again we see that smaller $\tau$ are more challenging for the m-hinge, but note how soft prices help smooth the (non-linear) loss curve and reducing the number of sharp local minima.

Interestingly, even when the class gap is zero or negative (left)—i.e., the positive class lies mostly to the left of the negative class—the market mechanism enables to obtain low loss values.[8] Here overall behavior is smoother for both the 0-1 loss and the m-hinge, due to smooth behavior of prices and price setters.

### B.2. Budgets and labels

Our empirical analysis in Sec. 3 focused on cases where budgets correlate with labels for a given $h$. While our goal there was to reveal how the strength and types of correlation can affect market outcomes, it does not imply that learning will always tend towards classifiers $h$ in which budgets and labels correlate along the direction that $h$ induces (or more precisely, on distances of negatively-classified points to the decision boundary). Importantly, we note that what matter is not the correlation between labels and budgets per se, but rather, the relation between labels and the normalized demand—which results from how budgets 'morph' distances to the decision boundary of a given $h$.

Here we demonstrate market outcomes on an example in which there is no correlation between $b$ and $y$. As we will see, what drives the classifier is user features in a way that circumvents dependence on budgets—which in this case, enables higher accuracy.

Our construction is as follows. Consider features in $\mathbb{R}^2$, and denote $x = (x_1, x_2)$. Points are generated such that each coordinate is sampled independently. For all points (regardless of class) $x_1$ is drawn from the distribution $\mathcal{N}(0, 0.4)$. For positive points $x_2$ is drawn from $\mathcal{N}(1, 0.3)$, and for negative points $x_2$ was drawn from $\mathcal{N}(-1, 0.3)$. Note the means of these distributions match labels, $\mu = 2y - 1$. The base rate is set to $p(y = 1) = 0.25$. Budgets were determined as:

$$b(x) = \max(0.1,\ 2.5 \cdot x_1 + \epsilon), \quad \epsilon \sim \mathcal{N}(0, 0.2).$$

i.e., budgets generally increase with $x_1$, but are noisy, and from above at capped at 0.1. Note that by construction:

- $b$ is correlated with $x_1$, but independent of $x_2$.

- $y$ is correlated with $x_2$, but independent of $x_1$.

We compare two threshold classifiers: $h_1$ which makes use only of $x_1$, and $h_2$ which makes use only of $x_2$. Each was trained on its respective feature to attain the maximal strategic accuracy on its induced market. Results are shown in Fig. 14:

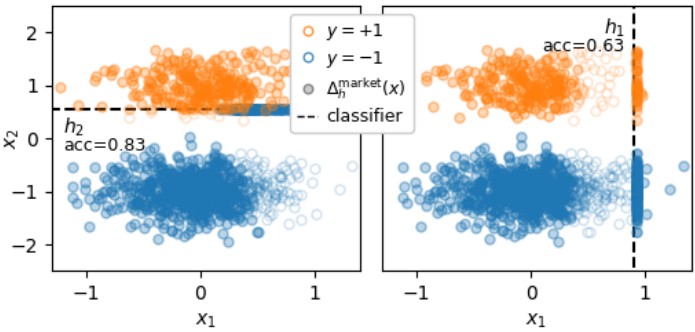

Figure 14: Revenue and prices for increasing sample size.

---

[8]This is surprising because our model class here includes only one-sided thresholds, $h_\tau(x) = \mathbb{1}\{x \geq \tau\}$.

As can be seen, the classifier $h_2$ (which uses $x_2$; left), yields better accuracy than $h_1$ (which uses $x_1$; right). Crucially, $h_2$ uses only $x_2$, which is independent of $b$, and so the classifier does not rely on any correlations between labels and budgets (via distances). Accuracy here is high since positive points are on the positive side of $h_2$, and the market enables only a subset of negative points to cross. Note that errors are precisely due to those negative points which do move: these are points with high $x_1$ values, and therefore those with higher budgets that permit cost-effective strategic movement. Conversely, accuracy for $h_1$ is low because it cannot limit only negative points from crossing. This is because the distribution of budgets (and distances) is the same for both classes for any choice of threshold.

### B.3. The cluster hypothesis

Our results in Sec. 5 demonstrated how for well-behaved demand distributions the price setter tends to be an extreme point. This was complemented by our theoretical characterization in Appendix A.3. Our conjecture is that the phenomena is broader, in that when the demand distribution comprises several "clusters", then the price setter will tend to be an extreme point of one of those clusters. Fig. 15 shows this holds for various mixtures of two Gaussians. We vary the relations between the Gaussians along several dimensions, such as distance from the decision boundary, gap between means, and variance. For all considered settings, results show that (i) the price setter is indeed an (almost) extreme point of one of the clusters, and (ii) the price setter "jumps" between clusters at particular points as we vary the parameters of the setup.

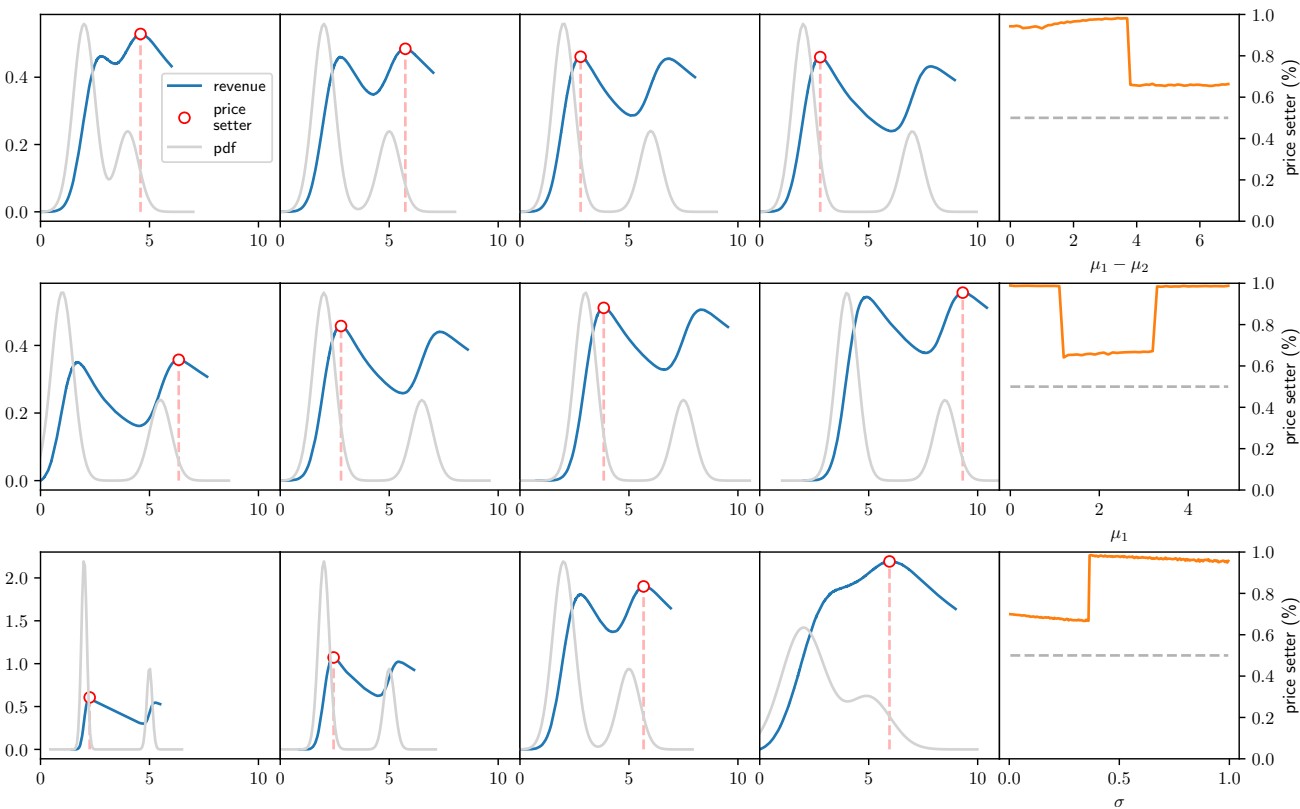

Figure 15: **The cluster hypothesis.** We conjecture that when the demand distribution is a mixture of well-behaved "clustered" distribution, the price setter will be an extreme point of one of the clusters. Varying the relations between clusters causes the price setter to "jump" from one cluster to the other. Here we vary: the gap between means of the two components **(top)**, the distance of the entire distribution from the decision boundary **(middle)**, and the variance of each component **(bottom)**. In each row, the four left plots show example distributions, while the rightmost plot shows the price setter (in percentage from the sample) for the entire parameter range. Note how each variation shows distinct jumps across clusters.

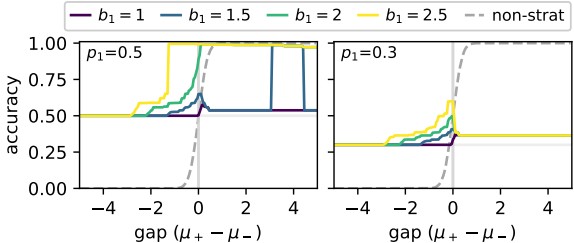

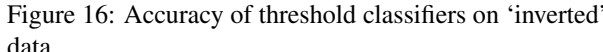

Figure 16: Accuracy of threshold classifiers on 'inverted' data.

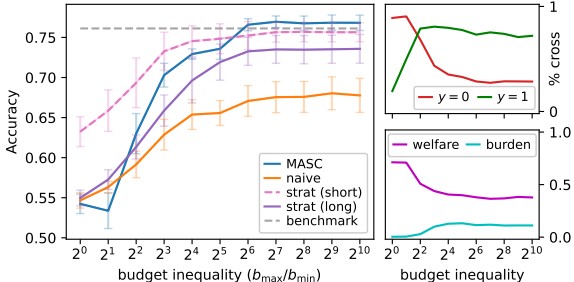

Figure 17: Results for the `folktables` dataset.

### B.4. Markets induce separability (further exploration)

Our results in Sec. 5.3 revealed a surprising result: that induced markets can make unseparable data become perfectly separable. Here we show that this effect can be even more extreme and unintuitive. Consider a univariate mixture distribution with class-conditional distributions $p(x \mid y) = \mathcal{N}(\mu_y, \sigma)$. We set $\sigma = 0.15$ and will be interested in the effects of varying $\mu$. We will examine both balanced data ($p_1 = p(y = 1)$) and class-imbalanced data with $p_1 = p(y = 1) = 0.3$. For budgets, we set $b = b_1 y$ and show results for $b_1 \in \{1, 1.5, 2, 2.5\}$. Our model class will consist of threshold functions $h_\tau(x) = \mathbb{1}\{x > \tau\}$. Note that we intentionally consider only thresholds that are oriented to classify larger $x$ as positive (i.e., the class does not include 'reverse' thresholds $\mathbb{1}\{x < \tau\}$).

Fig. 16 shows accuracies for the optimal threshold classifier across increasing gaps between class-conditional means $\mu_1 - \mu_0$. Note that a negative gap means that the positive distribution generates values that are mostly *smaller* than the negative distribution. The plot shows results for the range of budget scales $b_1$, and plot the performance of a non-strategic benchmark (dashed grey). As expected, the benchmark attains reasonable accuracy when the gap is large, provides $p_1$ accuracy when the gap is zero (and the two distributions are superimposed), and deteriorates quickly as the gap becomes more negative. But for market-aware classifiers, this is not the case. For $p_1 = 0.5$ (left), we see that for the larger budgets, accuracy can be 1 *even when the gap is negative*. For smaller budgets, outcomes under a negative gap can be *better* than under a positive! This latter behavior is much more distinct for $p_1 = 0.3$ (right), where for all budget considered, a positive gap enables significantly lower accuracy than moderate negative gaps allow.

### B.5. Main experiment: results on `folktables` dataset

Here we reproduce our main experiment from Sec. 6 on an additional dataset, building on the `folktables` data due to Ding et al. (2021). For the target variable $y$ we use a binarized version of the 'employment status' feature (code ESR), budgets we use the 'total income' feature (code PINCP). Appendix D.1.2 includes further details on the data and preprocessing.

Fig. 17 shows the results. Overall trends are similar to those of `adult` in Fig. 7 from Sec. 6, though there are several notable distinctions. As in `adult`, here as well all methods improve with scale, and our `MASC` approach outperforms others—here from scale $\alpha = 2^2$ and higher. Note that improvement for `naïve` is more pronounced. One key different is that the benchmark is not only much higher, but also hard to improve upon even for `MASC`. This can be explained by the lower ratio of positive crossers, i.e., even at large scales it is hard to allow positives to cross while preventing negatives (top right). Another difference is that `strat` underperforms by a larger margin and across all scales. Note that there is also a larger and consistent gap between `short` and `long` performance, suggesting stronger market adaptation.

## C. Differentiable market prices

Our market-aware learning approach replaces exact market prices $\rho^*$ with a smooth surrogate $\tilde{\rho}$ to enable differentiation. This is achieved by modifying the exact pricing scheme in Algorithm 1 to be differentiable.

One useful property of Algorithm 1 is that each of its steps can be easily vectorized, and each atomic operation is either already differentiable, or can be made differentiable using existing smoothing methods. In particular, note that: (i) $\text{dist}^+$ is a differentiable operator; (ii) `sort` can be implemented as a linear operator $\Pi$ with $\Pi_{ij} = 1$ if item $i$ is in position $j$ and 0 otherwise; and (iii) $U$ can be computed using a cumulative sum, implemented as linear operator $C$ with entries $C_{ij} = \mathbb{1}\{i \leq j\}$. The remaining non-differentiable operations are $\Pi$ and the argmax for choosing the revenue-maximizing

point $i^*$. Thus, if we replace $\Pi$ with a differentiable softsort operator $\widetilde{\Pi}$ (e.g., using Prillo & Eisenschlos (2020)) and the argmax with a softmax, then the entire algorithm becomes differentiable. These steps comprise Algorithm 2, which returns smoothed market prices $\tilde{\rho}$ as an approximation to $\hat{\rho}$. The final differentiable market price vector can be obtained as $\tilde{\boldsymbol{p}} = \tilde{\rho}w$, but is unnecessary to compute when using our market hinge loss, which requires only $\tilde{\rho}$. Note both softsort and softmax operators require setting appropriate temperature parameters.

---

**Algorithm 2** Smoothed empirical market prices

---

1: **input:** unit-budget pairs $\{(u_i, b_i)\}_{i=1}^n$ with $u_i > 0 \, \forall i$
2: $\bar{\boldsymbol{u}} = (u_1/b_1, \ldots, u_n/b_n)$
3: $\widetilde{\Pi} \leftarrow \texttt{softsort}(\bar{u})$ ▷ *approx. sorting matrix*
4: $\boldsymbol{u}_{\tilde{\Pi}} \leftarrow \widetilde{\Pi}\boldsymbol{u}, \;\; \bar{\boldsymbol{u}}_{\tilde{\Pi}} \leftarrow \widetilde{\Pi}\bar{\boldsymbol{u}}$
5: $\gamma \leftarrow \min \bar{u}$
6: $\bar{u} \leftarrow \bar{u}/\gamma$ ▷ *normalize demand*
7: $z \leftarrow 1/\bar{\boldsymbol{u}}_{\tilde{\Pi}}$
8: $\boldsymbol{c} \leftarrow \texttt{cumsum}(\boldsymbol{u}_{\tilde{\Pi}})$
9: $\boldsymbol{r} \leftarrow z^\top \boldsymbol{c}$
10: $\tilde{\rho} = z^\top \texttt{softmax}(\boldsymbol{r}) \cdot \gamma$ ▷ *de-normalize prices*
11: **return:** $\tilde{\rho}$

---

**Normalization.** In practice, we found it useful to normalize $\bar{\boldsymbol{u}}$ so that its smallest entry is 1. This is possible since market prices are insensitive to scale: if $\hat{\rho}$ is optimal for $\bar{\boldsymbol{u}}$, then for a scaled $\alpha\bar{\boldsymbol{u}}$, the solution is $\frac{1}{\alpha}\hat{\rho}$. Normalizing ensures that all temperature parameters (e.g., in softsort and softmax) operate at the same scale across all batches, which is important since the relation $\rho = b/u$ suggests that even mild perturbations to small $u$-s can cause large variation in computed prices.

**Truncated demand.** Recall that demand is determined by the distances of all points the lie on the negative side of $h$. In principle, since points on the positive side are assigned $u = 0$, their presence does not affect prices. However, when using soft prices, this does have a mild effect. To see this, consider that softsort employs row-wise softmax operations that replace the argmax used to indicate the sorting position. Since scores for all entries are exponentiated, points with $u = 0$ now contribute $e^0 = 1$ to the denominator. This biases outcomes, and becomes significant when there are many positively classified points. We circumvent this problem by simply truncating all points with $u = 0$ completely from the calculation.

**Hyper-parameters.** The smoothness of prices can be adjusting via two temperature hyper-parameters: $T_{\text{softsort}}$ for the soft sort operator, and $T_{\text{softmax}}$ for the final softmax. In the limit, these recover the exact argsort and argmax operators, respectively. Varying these temperatures therefore trades off approximation and smoothness (for optimization purposes).

# D. Experimental details

## D.1. Data and preprocessing

### D.1.1. ADULT

**Data description.** Our main experiment makes use of the `adult` dataset. This dataset contains features based on census data from the 1994 census database that describe demographic and financial data. There are 14 features, 8 of which are categorical and the others numerical. The binary label is whether a person's income exceeds \$50k. The dataset includes a total of 48,842 entries, 76% of which are labeled as negative. The data is publicly available at https://archive.ics.uci.edu/dataset/2/adult.

**Preprocessing.** To make the data appropriate to our strategic market setting, we took the following steps. First, all rows with missing values were removed (7.4%). Two features were excluded: `native_country`, and `education`, which had perfect correlation with the numerical feature `education_num`. The feature `capital_gain` was not used as input to the classifier, but rather as the basis of determining budgets. Second, to maintain class balance, 25% of negative examples were randomly removed. Finally, because behavior in strategic classification applies to continuous features, for our main experiment we dropped all categorical features. These however were still used for constructing budgets (see below). The remaining numerical features were normalized.

**Budgets.** For budgets $b$, we chose to use the `capital_gain` feature; of all features, this most closely related to an indication

of wealth. Unfortunately, only 8.5% (3,561) of the entries in the data contained values that were not 0, 99999, or NaN. As such, we decided to replace such missing or extreme entries with imputed values, for which we trained two random forest models (one per class) on the valid subset of the data. Hyper-parameters for this process were chosen using a grid search with the following parameters: n_estimators $\in \{50, 100, 200\}$, max_depth $\in \{$None, 10, 20, 30$\}$, min_samples_split $\in \{2, 5, 10\}$, min_samples_leaf $\in \{1, 2, 4\}$, 5 folds, and $R^2$ scoring. All features were used during imputation. The normalized RMSE was 0.366 for positives and 0.679 for negatives.

**Data splits.** We used a train-validation-test split of 70:10:20 and averaged the results over 10 random data splits.

### D.1.2. FOLKTABLES

**Data description.** We reproduce the main experiment of Sec. 6 using folktables as an additional dataset (Ding et al., 2021)—see Appendix. B.5. The data is publicly available at https://github.com/socialfoundations/folktables.

**Features, budgets, and labels.** We made use of the following features: 'AGEP' – age (int), 'SCHL' – educational Attainment (ordinal 0-24), 'MAR' – marital statue (categorical encoded to 1-5), 'RELP' – religious Affiliation (categorical 1-7), 'ESP' – employment status of parents (categorical 0-7), 'CIT' – citizenship status (categorical 1-5). For the target variable $y$ we used 'ESR' – employment status, converted into binary "employed" and "not employed". For budgets $b$, we used 'PINCP' – total person's income (float).

**Preprocessing.** All features where scaled to $[0, 1]$. Features that were negatively correlated with $y$ were flipped as $x_i \mapsto (1 - x_i)$. Examples with extreme or outlier budget values (below 50 and above 50,000) where removed.

**Data splits.** Same as adult.

## D.2. Evaluation

**Metrics.** In addition to accuracy, we measure the following metrics:

- **Welfare:** Measures the profit (utility minus cost) for all users of the system as:

$$\texttt{welfare}(h) = \frac{1}{B} \sum_i b_i \mathbb{1}\{h(x_i^h) = 1\} - c(x_i, x_i^h) \tag{16}$$

- **Social burden:** Measures the overall cost required to ensure that all deserving users (i.e., with $y = 1$) rightfully obtain positive predictions ($\hat{y} = 1$):

$$\texttt{burden}(h) = \frac{1}{B_+} \sum_{i:y_i=1} \min_{x':h(x')=1} c(x_i, x') \tag{17}$$

Here $B = \sum_i b_i$ is the total budgets, and $B_+ = \sum_{i:y_i=1} b_i$ is the total budget of the positive examples. Since the different experimental settings vary considerably in the distribution of budgets as well as its total, normalizing by $B$ and $B_+$ permits meaningful comparisons across conditions.

## D.3. Training, tuning, and optimization

**Implementation.** All code was implemented in python, and the learning framework was implemented using Pytorch.

**Optimization.** Our overall approach is to optimize the objective in Eq. (13) using gradient methods. In particular, we use ADAM with mini-batch updates —see details below. Additional decisions and considerations:

- The softsort and softmax hyper-parameters are intended to facilitate differentiable prices. In general, tuning their parameters should seek to optimally trade off between how well they approximate 'hard' sort and argmax, and the effectiveness of gradients. However, particular to our market settings, we observed that they also contribute to smoothing out discontinuities that result from sharp transitions between market states, i.e., cases where a mild change in prices causes a large change in the number of points that move and cross—which can significantly affect the loss.

- Similarly, we observed that mini-batches also have a smoothing effect on the market. This however related to a different aspect: Since market prices $\hat{\rho}$ correspond to the normalized demand of one of the data points $\bar{u}_i^{-1}$, prices in general

can be sensitive to the particular sample on which they are computed. Another concern if a small change in learned parameters move some points $x$ from being slightly above the decision boundary to slightly below it. If this occurs, then this new point has $u$ that is positive but very small, which can affect soft prices (despite our normalization step in Algorithm 2) through the choice of hyperparameters. Mini-batches help in this regard because they average out the effect that any single data point may have. They are also helpful in cases when several points 'compete' over setting the price (i.e., entail similarly large revenue) by permitting $\tilde{\rho}$ to express their (weighted) averaged.

**Initialization.** The model was initialized with the weights and bias term of the `naïve` model. Notably, initializing it with randomly generated weights from a normal distribution had minimal impact on the results.

**Hyperparameters.** We used the following hyperparameters:

- Temperature $T_{\text{softsort}}$ for the softsort operator: 0.001

- Temperature $T_{\text{softmax}}$ for the softmax operator: 0.01

- Batch size: 500

- Learning rate:

  - `adult`: 0.001 for `budget_scale` $\in [1, 32]$, 0.01 for `budget_scale` $\in [64, 1024]$
  - `folktables`: 0.001 for all budget scales

- Regularization and coefficient: 0.1

- Epochs: 100 for `adult`, 1000 for `folktables`

Hyperparameters were chosen by standard hyperparameter search over a grid of possible combinations and were chosen based on performance on a validation set along with considerations for reasonable convergance times.

