# OpenReview forum: "Learning Classifiers That Induce Markets"
_ICML.cc/2025/Conference — ICML 2025 poster_

### Official Review · Reviewer_Y1a5 · 2025-03-13

**Overall Recommendation:** 4

**Summary:**

This paper considers a standard binary strategic classification problem with a twist: the costs of manipulating features are endogenized, i.e., determined by a market. For example, college applicants could improve their SAT scores by paying for an SAT prep course, but the cost of the course is determined by market prices. The theoretical and empirical results focus on linear classifiers, linear costs for improving features and the utility of the decision-makers is their classification accuracy. They provide an algorithm for computing market prices for feature improvements given demand for features, and formulate a differentiable proxy objective for learning a classifier. Then they explore the range of outcomes that can occur using simulations, including a simulation calibrated to the Adult dataset.

## Update after rebuttal: I still view the setting and results as interesting. The clarity and questions of the other reviewers does not make me doubt this. I will keep my scores.

**Claims And Evidence:**

Overall, the idea of endogenizing manipulation costs is an interesting one, and I found the questions answered in the paper to be interesting.

Several aspects of the model formulation (mostly simplifying assumptions) should have been more clearly justified. For example, the use of linear classifiers: Either (1) Linear classifiers are commonly deployed in the motivating examples the authors consider. (2) Linear classifiers make the problem tractable and the insights generated are interesting. In either case, beyond explaining why the authors made this assumption, I would have liked to see the analysis extended to other settings (even if just via simulations) to see how the insights change under different analysis choices. I have similar questions about the cost for manipulating features.

Similarly, there was no analysis or comparison of the surrogate loss function or why it is worth including, beyond the property that it is differentiable and so amenable to gradient-based methods. There was no comparison between solutions obtained by optimizing for the surrogate loss and the global minimum, either theoretically or empirically.

One of the main qualitative claims in the paper is that in the market setting, most individuals are able to attain the positive classification. This is surprising, and makes me doubt how well the model is describing real-world strategic classification contexts. In applications of interest, do we see this behavior? If not, my guess is that this result comes from the zero capacity constraints and zero production costs, and don’t fit well with real-world contexts of interest. It would be interesting to consider relaxations of these assumptions, and the authors should further justify why they decided to build in these assumptions and why they are illuminating.

**Essential References Not Discussed:**

N/A

**Experimental Designs Or Analyses:**

The simulations were very interesting, well-motivated and yielded interesting conclusions!

**Methods And Evaluation Criteria:**

The theoretical results were limited (see the lack of evaluation of the surrogate loss above), and the empirical examples mostly relied on toy examples. I found the simulations to be fairly comprehensive and illuminating about the different possible outcomes. I really appreciated the construction of a classifier that is completely orthogonal to the feature that would be relevant in a non-strategic setting. Generally, the takeaway that strategic classification may lead to classifiers that behave totally differently than naive ones is interesting and worth further exploration. I also found the observations that if budgets correlate with distance to the decision boundary, it is possible to set classification thresholds so as to get high accuracy to be interesting in the simulations. In future work, I would like to see this proved formally, rather than in simulations.

I didn’t understand the connection between Theorem 2 and the fact that almost all points cross, and there wasn't enough discussion for me to follow it.

**Other Comments Or Suggestions:**

N/A

**Other Strengths And Weaknesses:**

The writing was clear and the problem is well-motivated. I wish the authors had spent more time on the observations in section 5 since these were more counterintuitive and interesting than sections 3 and 4.

**Questions For Authors:**

Why is the surrogate loss good? Does it have good properties? Did you check performance of optimizing against the surrogate loss against brute force solutions.

**Relation To Broader Scientific Literature:**

This paper contributes a novel model of strategic classification (where costs to change features are endogenous). To my knowledge, this is novel.

**Theoretical Claims:**

Proof of theorem 1 looks good to me. I didn't check the ones in the appendix.

The techniques used mainly involve solving linear programs and analyzing the simple game induced between the decision-maker and decision subjects. I didn't find the theoretical results very compelling, especially since they strongly depend on the linearity assumptions and structure of price setting on the part of sellers.

---

> ### Author Rebuttal · Authors · 2025-03-31
>
> Thank you for your encouraging review and insightful questions.
>
> > Simplifying assumptions such as linear classifiers and linear costs
>
> Our choice to focus on a simple setup stems from several considerations. Indeed, one consideration is tractability (of both pricing and learning problems). Another consideration is simplicity as a guiding principle: as a first step to exploring market-inducing classification, we believe our choices are reasonable. Note also that almost all works on strategic classification focus on linear models and simple costs (e.g., linear, 2-norm). Since our formalism layers on an additional market mechanism, preserving this structure seemed useful. Finally, we hope you agree that despite our simplifying assumptions – the phenomena that arise from the market mechanism are sufficiently interesting, and even surprising, to merit simplifying assumptions. That said – we certainly agree that more elaborate settings are worth pursuing in future work.
>
> > No analysis or comparison of the surrogate loss function or why it is worth including
>
> This is a great suggestion! **We will gladly add a comparison of the 0-1 loss vs. our m-hinge loss** for settings where computing the 0-1 loss is tractable. In fact, we already have such results for some settings; for example, in Fig. 4 (which shows 0-1 accuracy), the hinge loss can be shown to replace “flat” regions with linear slopes – as can be expected. This suggests that the hinge serves as a useful proxy when slopes push the model towards “lower” flat regions. We can add this to the figure (it was actually removed due to clutter), and include further analysis on other examples in the Appendix.
>
> > In the market setting, most individuals are able to attain the positive classification
>
> We think we understand your concern, but this claim is not entirely precise. Looking at the results in Fig. 2, it is true that most points end up being able to cross. Fig. 3 suggests that this effect reduces when budgets are more diverse. But both examples consider a single “cluster” of points; once there are more clusters, then it is no longer the case that all or most points will move. The more general phenomena we believe is at play is that *clusters move together*, i.e., the price setter will tend to be an extreme point of *some* cluster. This can be seen for example in Fig. 4 (bottom right): note how changing the budget ratio (y-axis) causes the price setter to jump from being the extreme point of one cluster to that of another.
> Given your input, we think it will be valuable to add further empirical investigation of this phenomena. We already have some initial results on this and will gladly add them, either in the Appendix or using the extra page of the final version.
>
> > How well [does] the model describe real-world strategic classification contexts?
>
> This is a good question which we feel summarizes well many of the previous comments. Real world market dynamics are very likely more complex than our model posits, especially if the market forms in response to a classifier. We believe our model, albeit its simplicity, is still able to capture (at least coarsely) certain effects of transitioning from fixed costs to those of an induced market. But of course this is speculation, and a definitive answer requires much further investigation and research efforts.
> In regards to production capacity and costs, these are certainly a natural next step to consider. One reason we focused on no constraints or costs is that this significantly reduced the number of free parameters required to specify the setup. Another subtler point is that it isn’t immediate how constraints and costs operate when transitioning from a finite sample (as in training) to expected outcomes (on which we aspire to evaluate). Having no constraints and costs makes it possible to have a single well-defined market mechanism that captures both and supports the notion of generalization.
>
> > I didn’t understand the connection between Theorem 2 and the fact that almost all points cross
>
> Thm. 2 states that, for the considered distributions, (i) there is a unique price setter, and (ii) it lies beyond the peak of uf(u). Intuitively, since this point is typically more extreme than the peak of f(u) itself, the price setter will be positioned at an extreme quantile. We will make this clearer in the next revision.

---

### Official Review · Reviewer_QMW2 · 2025-03-14

**Overall Recommendation:** 4

**Summary:**

This paper extends strategic classification to a setting where users seeking positive predictions can purchase features from sellers, leading to the formation of a competitive market. The authors analyze how users respond to prices, how market prices adjust based on demand, and how classifiers influence these dynamics. The authors propose an efficient algorithm for computing prices and introduce a learning framework that takes into account the market effects of a classifier. The authors also demonstrate how the market-aware strategic
learning framework performs empirically on real data with simulated market behavior.

**Claims And Evidence:**

Yes. The claims are well motivated, theoretically sound and supported with experiments.

**Essential References Not Discussed:**

References are adequate.

**Experimental Designs Or Analyses:**

Yes. They seem fine but would've preferred to validate on multiple datasets instead of just adult income.

**Methods And Evaluation Criteria:**

The proposed methods and evaluation criteria make sense

**Other Comments Or Suggestions:**

Minor typos here and there.
Eqn 8, typo in constraint
Ln 213: Aglorithm

**Other Strengths And Weaknesses:**

The paper is generally well-written with novel ideas that are theoretically sound, supported by a algorithms for computing prices and learning market-aware classifiers and decent validation through empirical experiments.

A potential weakness is in the simplifying assumptions (such as linearity in cost modeling etc). Another weakness is that the evaluation is quite limited to a single real-world dataset (Adult Income).

**Questions For Authors:**

Refer above to comment on datasets/ experiments.

**Relation To Broader Scientific Literature:**

This paper extends strategic classification and combines it with topics from the markets and learning literature. I found the direction exciting and quite novel. Further empirical validation on real-world economic datasets would strengthen the claim and relevance.

**Theoretical Claims:**

Yes (Generally went through the statements but didn't check too much in detail)

---

> ### Author Rebuttal · Authors · 2025-03-31
>
> Response:
>
> Thank you for your positive review! We were happy to hear that you found our paper exciting and novel. If you have any further questions we would be glad to discuss.
>
> > Would've preferred to validate on multiple datasets
>
> We are happy to report that **we have extended our experimental section to include an additional dataset**. For details please see our response to Rev. 8pCZ.
>
> > Simplifying assumptions such as linearity in cost modeling etc.
>
> We agree that supporting more complex costs and models would have been a nice addition. However, as a first step, we think that it is reasonable to focus on a linear construction – especially given that most works on strategic classification consider also linear classifiers and simple cost functions (e.g., linear, 2-norm, or squared).

---

### Official Review · Reviewer_8pCZ · 2025-03-14

**Overall Recommendation:** 2

**Summary:**

The paper studies strategic classification in settings where the cost function for modifying inputs depends on the chosen classifier, via the market this classifier induces. In particular, the chosen classifier determines which features are more "important" for positive decisions and therefore affects the demands for each feature, thus also impacting the market equilibrium. The authors propose a natural mathematical framework for this problem, extending the classic strategic classification model. Then, they derive the equilibrium prices for the case of linear classification and propose an algorithm for computing the empirical equilibrium prices. They propose a learning algorithm for their problem and evaluate it on the adult dataset, comparing it to existing methods. The also market adaptation to classifiers in several simple settings, providing further insights into their model.

**Claims And Evidence:**

The paper proposes a new framework for strategic classification, motivated by the observation that prices for features may depend on the classifier itself. I find the problem interesting and the proposed model natural and relevant.

Overall, I think that the authors derive an algorithm and conduct an evaluation which are reasonable. That said, several aspects of the work can be strengthened, to make the contribution more convincing.

In particular, the authors propose a surrogate hinge-like loss for their problem. However, in Figure 6 the proposed method underperforms compared to a standard strategic method. This brings the question of whether another loss surrogate can perform better - perhaps an ablation study for the role of the hinge loss can be helpful here?

Similarly, experiments on other datasets, e.g. Folktables or some synthetic example like in Section 5 can help to bring further evidence for the empirical effectiveness of the proposed algorithm.

**Essential References Not Discussed:**

NA

**Experimental Designs Or Analyses:**

See above.

**Methods And Evaluation Criteria:**

See above.

At a few places, the notation and assumptions remain a bit unclear to me - please see below at "Other Comments Or Suggestions" for a few specific suggestions and requests for clarifications.

**Other Comments Or Suggestions:**

- In equation 4, I guess delta also depends on b?
- The authors refer to the "demand set" before equation (5). It will be good to give a formal definition of this concept, or at least refer to some relevant source.
- At a few places, the authors refer to the linear models parameters as w and a, rather that w and tau - it will be nice to sync that throughout the text.
- In equation 6, Delta is defined as the amount of feature change, rather than the new resulting data point. However, in equation 7, it seems to be used as the resulting data point?
- Before equation 10, the authors state that they assume that "sellers have foresight". Could you please provide a technical statement for this assumption?

**Other Strengths And Weaknesses:**

NA

**Questions For Authors:**

See above.

**Relation To Broader Scientific Literature:**

The conceptual contribution of the paper is clear and extends the framework of strategic classification in a meaningful way.

The paper of Chen et al. (2024) is mentioned in the related work, however it's unclear to what extend their model and/or techniques are relevant here. Perhaps the authors can elaborate?

**Theoretical Claims:**

The derived theoretical results are interesting and interpretable.

In Section 2, the authors claim that the prices are positive. However, in Section 3 we see that the equilibrium prices are proportional to the linear weights. Am I missing something, or can the weights, in general, be both positive and/or negative? How is that compatible with the prices being constrained to be non-negative?

---

> ### Author Rebuttal · Authors · 2025-03-31
>
> Thank you for your review and comments. We were glad to hear you see our paper as making a clear conceptual contribution – this was indeed our primary aim and focus.
>
> Your review mentions that you believe our results can be strengthened, in particular by considering (i) alternative proxy losses and (ii) more datasets. In regards to (ii), and as per your suggestion, **we have extended our empirical results to include an additional dataset based on Folkstable**. Results here confirm most of our previous findings and provide additional insights. Regarding (i), we address this and all other points below.
>
> > The proposed method underperforms compared to a standard strategic method
>
> This is true, but only for small budget scales ($\le 8$). Note the original data has scale $2^{10}$ (star marker), for which our approach clearly outperforms the baselines and by a large margin. Note also that the x-axis is in logarithmic scale: our method is better in the range $[16,1024]$.
> In terms of results, we consider this phenomena an interesting finding. In contrast to our approach (MASC) which anticipates prices that *adapt* to the learned classifier (at equilibrium), the standard approach (strat) assumes fixed prices. Our interpretation of the results is that when the distribution of budgets approaches uniform, price adaptation becomes either mild or inconsequential. In this regime, the “price” we pay to enable differentiability turns out to be larger than the benefits of accounting for price equilibration.
>
> > Perhaps another loss surrogate can perform better?
>
> This is certainly possible, and we would love for future work to develop better solutions. Note however that designing proxy losses for strategic learning tasks can be quite challenging. Even for standard strategic classification, fundamental concepts such as margins can break completely (see Levanon & Rosenfeld (2022)). The only existing approaches that we are aware of and that apply to our setting are the s-hinge (which we build on) and Hardt et al. (2016), which underlies our baseline. For our market setting, even the connection to the s-hinge is not straightforward, as it is intended for the 2-norm cost – not linear, and the generic extension of the s-hinge to other costs is generally intractable.
>
> > Experiments on other datasets
>
> We are happy to report that our experimental section has been extended to include another dataset based on Folktables (thanks for this suggestion!). The target variable is employment status, and budgets derive from income. Results show overall similar trends to Adult, but with some distinctions. All methods improve as inequality increases. Our method outperforms all others – here at all scales $\ge 4$. The % of crosses, welfare, and social burden behave similarly to Adult. Interestingly, and in contrast to Adult, here strat *underperforms* across all scales, even when compared to naive. We will add these results to the final version.
>
> > Can the weights, in general, be both positive and/or negative?
>
> Although possible, our approach does not explicitly constrain weights to be positive. This is in line with the results of Hardt et al. (2016) for non-adaptive linear costs. We do however expect them to be such for the learned classifier (otherwise, users would be paying to decrease feature values).
>
> > Vs. Chen et al. (2024):
>
> This recent paper is similar to ours in that it generalizes strategic classification to support dependencies across user response through the cost function. The main difference is that in their setup, dependencies are encoded explicitly as externalities in the cost function, which is fixed and predetermined. This means that, as in standard strategic classification setup, learning requires knowledge of the particular cost function. In contrast, our work models dependencies as forming indirectly through the market mechanism; the cost function itself is adaptive, and learning only requires knowledge of how the market operates. Another distinction is that externalities depend on the classifier only indirectly: explicitly, they depend on how other points move. Market costs depend on demand rather than on actions; given prices, actions become independent. These distinctions make it hard to draw direct connections between the results and methods of our work and theirs.
>
> > Minors
>
> - Eq. (4): $\delta$ is defined as the amount purchased. This implicitly depends on b, which constrains how much *can* be purchased.
> - Demand set: Eq. (5) is the definition. Demand set in general is a common construct.
> - a vs. $\tau$: Thanks – we will correct this.
> - $\Delta$ in Eq. (6): This is true! Thank you for noticing. The fix is to add $x+$ before the argmax.
> - Sellers have foresight: See e.g. “Noncooperative Collusion under Imperfect Price Information” (Green & Porter, 1984).
>
> ----
>
> **Thank you again for your feedback. We hope our response and improvements meet your expectations, and kindly ask you to consider increasing your score.**
>
> ----

---

> > ### Comment · Reviewer_8pCZ · 2025-04-02
> >
> > I would like to thank the authors for their response. The folktable results look promising and I believe they should be included in the text, together with their clarifications.
> >
> > My main concern that remains is about the prices being positive/negative. In Section 2, the authors explicitly define the prices as non-negative values. However, in Section 3 onward, nothing in the proposed method seems to constraint the weights from becoming non-negative. Since the equilibrium prices are proven to be proportional to the weights (Proposition 1), this also implies that the prices can be negative.
> >
> > In the rebuttal the authors state that they do expect the weights (and prices) to be non-negative for the learned classifier. Is this actually the case in the experiments? This seems hard to ensure, as not all features in benchmark datasets will be positively correlated with the predicted outcome. Even from a theory standpoint, if all variables are (positively) predictive of a positive outcome, they may be correlated and an optimal classifier may assign negative weights to some perhaps?
> >
> > I will be grateful if the authors can elaborate more on this issue.
> >
> > Additionally, I believe that further details (and citations) on the notions of "demand set" and "seller foresight" should be provided, as I expect many readers of ICML papers to not be familiar with these terms.

---

> > > ### Author Response · Authors · 2025-04-06
> > >
> > > Thank you for your response! We will make sure to include the Folktables results in the paper, and properly define demand sets and seller foresight. Thank you again for both suggestions.
> > >
> > > As for prices – we will gladly elaborate here further. The answer is somewhat nuanced, so please allow us to clarify (and apologies in advance for the lengthy response).
> > >
> > > First, regarding experimental results: indeed the majority of our experiments resulted in classifiers with positive weights. Those that did not had only a few negative entries, and with small absolute values. In addition, we reran all experiments with an additional constraint enforcing $w \ge 0$ (implemented using projected gradient descent). Results for our method are virtually unchanged (up to noise from randomization).
> > >
> > > Second, and nonetheless, we agree that our method as currently presented does not explicitly enforce positive weights. One (easy) solution would be to add this constraint to the setup. This is certainly possible – and we would happily consider this if you believe it would make things clearer. But at the same time, we would like to stress that our method and results are sound even without this assumption.
> > > The reason is that even if some weights are negative, there still exists a price vector $p’ \ge 0$ such that (i) $p’$ is an equilibrium price, and (ii) outcomes are the same under $p=w$ and $p’$. This is because items are exchangeable: the price a single point x pays for crossing is the same regardless of which features are eventually bought. Hence, there exist many equilibrium prices (which is common in markets with exchangeable items). Our method makes use of the particular choice of $p=w$ since this enables us to adapt the s-hinge to our purposes (*). But as long as not all weights are negative, the market remains well-defined, because even if the particular equilibrium $p=w$ is not feasible, other $p’ \ge 0$ are. Similarly, our method remains effective because the set of points that move is the same under any equilibrium price (whether negative or positive), and so outcomes (and hence the loss and accuracy) are also the same. The proof for this is simple: it shows that regardless of which features are “bought”, demand remains similarly proportional, and the price setter is invariant to this choice. We will add the formal claim and a full proof to the Appendix.
> > >
> > > Finally, we note that it is generally possible to work with prices $p=w$ in which $w$ has negative entries if we interpret these as meaning that users need to “pay to get less” of something. For example, if a feature encodes weight, then we can pay the gym to reduce it; if a feature encodes the size of a house, then change in any direction is costly. This extends beyond the setting we present in the paper, but is still a feasible interpretation. Technically it requires constraining that $x^h$ remains positive, but this would not change our current results. We thought it would be clearer to focus on positive prices (despite the loss of generality). But if you feel otherwise, then we can certainly consider this alternative.
> > >
> > > (\*) The s-hinge itself is designed for L2 costs, and is inappropriate for fixed linear costs. Our construction works because, for our choice of $p^*$, points move “as if” towards the decision boundary – as they do under L2 costs. This made it possible to adapt the s-hinge to become our m-hinge, which works for linear *market* costs since they adapt to the classifier. It would have been equally possible to work with non-negative equilibrium prices, but this requires an additional normalizing constant, which we preferred to sidestep.

---

### Official Review · Reviewer_XboL · 2025-03-18

**Overall Recommendation:** 2

**Summary:**

The authors propose a market-based perspective in strategic classification and challenge the key assumption that cost functions do not depend on the classifier and are fixed. The paper builds on the premise that classifiers, when used in the real world, incur demand for their features, especially when they lead to a desired prediction. To this end, the authors present a proof of concept using linear classifiers and conduct an empirical study on the `adult` dataset to demonstrate how classifiers impact markets and vice versa.

**Claims And Evidence:**

> In standard strategic classification, a useful strategy that exploits this idea is to 'raise the bar'... (line 363)

Is there citation for this?

**Essential References Not Discussed:**

I couldn't think of essential references that the authors did not mention.

**Experimental Designs Or Analyses:**

The experimental design itself doesn't have major flaws. Perhaps one could complain about the lack of other datasets, but as long as there are meaningful results and analysis, the number of datasets is not a big issue.

However, this paper's empirical results analysis is woefully insufficient. The results section merely describes what we see in Figure 6 and does not derive any meaning from it. I think there is a missed opportunity with the *burden* metric; when I read section 5.3 and Figure 3, I immediately thought of the cost of such classifiers for those with $y = 1$ -- there is quite a bit to discuss here.

**Methods And Evaluation Criteria:**

No issues here.

**Other Comments Or Suggestions:**

- line 296: should be right-skew?
- line 303: $b_\textrm{max} = 2^{-\alpha}$: isn't this smaller than $b_\textrm{min} = 1$ for $\alpha > 0$ ?
- Figure 5: would appreciate if there could be a better way to indicate $\Delta_{h}(x)$? Its very hard to differentiate the points.
- If the authors want to use the term "negative" outcome, they may want to switch to $y\in \\{-1,1\\}$. There are some typos regarding classes (e.g., eq (8), line 193)

**Other Strengths And Weaknesses:**

Three major weaknesses:
1. **Exposition**
Although the technical details are not widely complex (which is fine), the paper is hard to follow. For example, in the introduction, there is an attempt to summarize the contributions of the work towards the end, but the takeaways are not clear.
> we show that markets can give rise to complex behavioral patterns that differ significantly from the conventional model of strategic classification (lines 85-88).

What are the behavioral patterns in conventional strategic classification? I think this is implied in the sentences that follow, but stating it explicitly and contrasting the findings would serve the paper better.

2. **Figures/Plots**
Another major issue (related to 1.) is how the plots are explained (or not). It takes a very long time to parse the figures (especially figures 2, 3, and 4). The captions are not descriptive. For example, I believe the revenue curves are drawn from a large sample size (since they are smooth), but I can't find any mention of that in the caption or the text. This was especially confusing since figure 1 shows how revenue curves are piecewise linear functions. Considering how figures are supposed to aid understanding, this is a major concern.

3. **Contribution**
Lastly, I am not convinced that this is a valuable contribution to strategic classification. This is perhaps an artifact of the lack of analysis in the results section, but I am unsure of the value of considering a classifier-dependent cost function through the paper's mechanism. If the performance is dependent on budgets, isn't the classifier really classifying based on the budget?

Furthermore, consider the last couple of sentences in the introduction:
> classifiers will be accurate under the markets if they induce only if they associate positive predictions with high budgets. This raises natural questions regarding fairness and socioeconomic equity.

This yields two questions:
-  How does this algorithm perform when budgets are not correlated with the outcome?
-  Do we want to build such classifiers? Or are the authors suggesting that this is an accurate description of reality?

**Questions For Authors:**

- Why the term "inequity" (vs. let's say inequality) in figure 3?

> setting $w=\boldsymbol{p}$  (line 411)
- Can the authors reason in detail how this draws on the idea of the main algorithm in [1]?

[1] Hardt, Moritz, et al. "Strategic classification." 2016.

**Relation To Broader Scientific Literature:**

See other strengths and weaknesses.

**Theoretical Claims:**

I didn't find major issues with the correctness of the paper's theoretical results. However, there were some incorrect claims that might have been a result of typos. See Questions.

---

> ### Author Rebuttal · Authors · 2025-03-31
>
> Thank you for your careful reading and important comments. Overall, it seems that your concerns are: (i) clarity of exposition and captions, (ii) discussion of the empirical findings in Sec. 6, (iii) the use of a single dataset, and (iv) contribution.
> For (i) and (ii), **we believe these are easily fixable**, especially given the extra page for the final version. For (iii), **we have added another dataset**, please see our response to Rev. 8pCZ. For (iv), **we believe our response below will help establish the contrary** by clarifying the role of budgets. In this case, we kindly hope you would be willing to reconsider your evaluation.
>
> > Empirical results analysis is insufficient
>
> Thank you for pointing this out. In hindsight we agree that a more thorough discussion of the results in Sec. 6 would be beneficial – we will gladly expand the discussion here. As you note, there are many potential insights to discuss within the existing results.
> Please note however that our empirical analysis spans both Sec. 5 (synthetic data) *and* Sec. 6 (real data). We hope you agree that our discussion in Sec. 5 is adequately comprehensive (as other reviewers note), and that together they provide a useful picture.
>
> > Exposition
>
> As per your suggestion, **we will add to the introduction a succinct summary of the paper’s contributions as bullet points**. We will also make the distinction from standard strategic classification clear and explicit.
> On a more general note: Our paper lies at the intersection of two disciplines. Presenting our setup and motivation in a way that is equally readable to both audiences is not an easy task. We have given this much thought, but it is possible that the current version leans more towards one than the other (given that other reviewers were happy with clarity). If there are any particular clarity issues you feel are present, please let us know and we will improve them.
>
> > Figures/plots
>
> Thank you for this input. We will gladly make use of the extra page to clarify and further elaborate.
>
> > If the performance is dependent on budgets, isn't the classifier really classifying based on the budget?
>
> **Generally, no.** Budgets are certainly important, and can play a significant role in determining learning outcomes, but are not the only component that matters. If this were the case, then learning itself would become degenerate. Our results indicate that this is not what happens. The relation between budgets and labels affects accuracy through the market mechanism – **but the market itself depends on the classifier**. Since the classifier h(x) does not take budgets as input, the relations between features x and labels y is what determines the space of *possible* markets. Learning *can* come to rely on features for which budget disparities are dominant (if exist), but it is equally plausible that learning will come to *avoid* such markets. We have included in the appendix a simple 2D example which illustrates this behavior. In the construction, b correlates with y in the direction of x1, but the optimal classifier uses only x2, under which the market is stagnant.
>
> > Last sentences in intro:
>
> Please note that these should be read in the context of the preceding sentence – not as a general independent claim.
>
> > How does this algorithm perform when budgets are not correlated with the outcome?
>
> This is a good question. In general, standard (linear) classifiers work well when the labels correlate with distances from the decision boundary. In strategic market settings, budgets can improve (or degrade) this correlation–which is why they can be helpful. Thus, what matters is the relation between budgets and labels **along the direction of the classifier** (or more precisely, the distances on its negative side). This depends on the classifier and is not an a-priori property of the data. This is also why learning should make use of features (see our point above).
> If budgets and labels are completely independent, then the choice of classifier should have little effect on the market. But the more likely scenario is that in some directions features will be more informative, and in others, budgets. An optimal solution is likely one that exploits the combination of these two different sources of information through how they form a market.
>
> > Do we want to build such classifiers? Or [is this] an accurate description of reality?
>
> The question of what we should “want” is of course multifaceted and depends on context. What we believe our results convey is that: (a) if we learn naively (in a way that does not account for the market), then market forces will obscure our performance, whereas (b) if we learn in a way that accounts for the market, then this can improve performance, but possibly by exploiting financial inequalities.
>
> > Minors:
>
> line 363: See Hardt et al. (2016).
> line 296: Yes, we will correct this.
> line 303: bmin and bmax have been swapped.
> Fig 5: We will add arrows for a subset of points.

---

> > ### Comment · Reviewer_XboL · 2025-04-04
> >
> > I thank the authors for their response.
> >
> > **Concerns raised by other reviewers**
> >
> > I know some reviewers have comments on the somewhat simple setting (i.e., restricting to a linear model, assuming $\delta_ \geq 0$). I understand that the authors did this to make theoretical analysis more tractable; I don't believe it is a major issue that impacts the contribution of the paper -- especially considering the authors are presenting a novel concept
> >
> > **Budgets**
> >
> > Having said that, my question/concern regarding budgets remain.
> >
> > I realized that my comment on whether the classifier is discriminating on the budget was not very clear. From what I understand:
> > - Learning procedure considers $x^h = \Delta_{h}^{\mathrm{market}}(x;b)$ (equation 13)
> > - Hence for a fixed classifier, my intuition is that the "market induced" $x^h$ depends on the budget $b$
> > - As a result, the learned classifier $h$ has internalized the correlation between budgets and labels (if it is useful for prediction, that is)
> > The authors seem to acknowledge in both the paper and the rebuttal that performance (with strategic agents) depends on how budgets are distributed across labels. For example, if individuals with positive labels have large budgets, this allows the learned classifier to outperform standard linear models. This is because the budgets allow the data post-strategic response to be more separable.
> >
> > With this in mind, I am confused as to why the authors say:
> > > what matters is the relation between budgets and labels **along the direction of the classifier**
> >
> > > Learning _can_ come to rely on features for which budget disparities are dominant
> >
> > I was under the assumption that budgets were universal (i.e., $b \in \mathbb{R}_+$ shared among all features).
> >
> > Still, it seems like the utility of this concept (which I admit is interesting) relies on the distribution of budgets.
> >
> > **Minor Additional Questions**
> > - Is $h_{\mathrm{market}}$ an output of the algorithm?
> >
> > ---
> > # Updated Rebuttal Response
> >
> > I thank the authors for their response and for clarifying the relationship between budgets, labels, and $x^h$.
> >
> > > Since h is a function of x alone, a classifier cannot generally “internalize” the relation between b and y because this has to go through x.
> >
> > Perhaps "internalize" was not the best word choice, but
> > >"only directions which induce markets where _moving_ correlates with labels can be helpful for accuracy"
> >
> > seems to suggest that an accurate model will use (which, I believe is the main point of the method) the relationship between the budget and label; this is because budgets determine (jointly) which points move.
> >
> > Which begs the question:
> >
> > **How does standard strategic classification perform in the example setting above? (i.e., when $b$ and $y$ are not correlated)** -- my intuition tells me that it might perform comparably.
> >
> > Furthermore, I want to ask the authors on how one should implement budgets in real-life applications; would it be disposable income (or bounded by it)? I agree that how cost functions are defined in current strategic classification literature can be unrealistic. While I commend the work for challenging the assumption that cost functions are predetermined and bringing it into discussion, I want to highlight that costs are often not uniform. Given an action (i.e., to purchase a feature) may have different costs for different individuals. I believe this is one of the reasons why modeling cost functions is very difficult.
> >
> > Nonetheless, I appreciate the authors' response and have raised my score.

---

> > > ### Author Response · Authors · 2025-04-06
> > >
> > > Thank you for your follow-up! We appreciate your willingness to further discuss these points with us. We hope our following response will help clarify the remaining issues.
> > >
> > > > Budgets
> > >
> > > Indeed, induced markets depend on the distribution of budgets (to us, this is what makes them interesting!). But the *direction* induced by w is crucial to determine *how* (and if) budgets affect outcomes. We are not entirely sure what remains unclear, but please allow us to try and shed light on some points that may be helpful:
> > > 1. While best responses $x^h$ do depend on budgets, please note that each $x_i^h$ depends on *all* budgets $b_1,\dots,b_m$, not only on its own $b_i$. This is implicit in the dependence on $p^h$ which is a function of the set $\\{(u_i,b_i)\\}$. It is therefore not true that if b correlates with y, points with larger b will simply “move further” and therefore improve accuracy. This is since the market coordinates movements across users, i.e., the $x_i^h$ are all dependent.
> > > 2. One way to see the role of directionality is through the observation that prices do not depend on budgets directly, but rather, on how they “morph” demand u – i.e., the directional distances to h; see the definition of units-per-budget $\bar{u}$ in line 215 (right) and their usage in step 5 in Algo. 1. This morphing can either be helpful (if it disentangles labels) or obstructive (if it mixes them). Both can happen even if b correlates with y.
> > > 3. Since h is a function of x alone, a classifier cannot generally “internalize” the relation between b and y because this has to go through x. The relation between x and b affects which points *move* (jointly, through the market); the relation between x and y affects accuracy. Hence, by conditioning on x, only directions which induce markets where *moving* correlates with labels can be helpful for accuracy.
> > >
> > > **Example.**
> > > To illustrate further, consider a simple 2D example (similar to that of Fig. 5) where x1,x2 are independent, and y is a function of x1. Now fix b to be an increasing function of x2. Note this means that **b and y are *not* correlated**, since (i) y depends only on x1, (ii) b depends only on x2, and (iii) x1 and x2 are independent.
> > > * Consider a classifier h2 that uses only x2. Since x2 correlates with b, the market will cause some points to cross (pending on the choice of threshold). But since x2 is uninformative of y, such movements are entirely unhelpful for accuracy; that is, points with y=0 and points with y=1 generally “move” together, and so **the market is unable to separate points by label**. The optimal h2 (threshold at ~1) attains accuracy of 0.63.
> > > * Now consider a classifier h1 that uses only x1. Since b is unrelated to x1, all points have the same p(b)=p(b|x1). Intuitively, this means that **the market is unable to separate points by their budgets**, as each interval of x1 has the same mix of budgets (which are spread uniformly along x1). Luckily, x1 *is* informative of y. The optimal h1 thresholds at ~0.5 (i.e., between the Gaussians), and only the portion of the negative points with higher budgets cross. Accuray here is 0.83 – and is optimal.
> > >
> > > Note that these results are the opposite of the original example in Fig. 5. This is precisely because in the above example b correlates with features, not labels. The example in Fig. 5 shows how when b and y start out correlated, the optimal classifier can “decorrelate” them. Our example here shows the opposite: the optimal classifier is completely unable to (nor needs to) discern between low-budget and high-budget users.
> > >
> > > **High-level rationale.**
> > > The standard strategic classification setting considers costs that are fixed and uniform across users. A primary goal of our work was to extend beyond this. The first step was to introduce a market mechanism: this makes costs adapt to the choice of classifier. But our results show that markets under uniform costs are typically “extreme” in that either none or (almost) all users move. This motivated allowing for individual budgets to vary, which we think is a plausible modeling choice.
> > > It is true that users with bigger budgets can generally “move further”; this is innate in the construction. But the fact that strategic learning tends to produce classifiers that discriminate on the basis of budgets, as our results suggest, is an emergent phenomena – not an immediate implication of the setup. Markets coordinate the behavior of many users in a non-trivial manner and introduce complex dependencies in who moves and who doesn’t. In our minds, the fact that learning can exacerbate budget inequalities *through* the market mechanism is an interesting finding.
> > >
> > > > Is $h_{market}$ an output of the algorithm?
> > >
> > > Yes, of the learning algorithm that accounts for strategic market behavior.

---

### Decision · Program_Chairs · 2025-05-01

**Decision:**

Accept (poster)

**Comment:**

The paper studies strategic classification where the cost of manipulation depends on the classifier.  On the positive side, the reviewers find the paper technically sound, the model natural, and the findings overall interesting.  On the negative side, there were questions about the significance of the contributions, as well as certain aspects of the experimental evaluation.  Overall the reviews collectively suggest this is a valuable contribution that could still be improved, which makes it a borderline case.  We encourage the authors to take the chance and further strengthen their paper regardless of the outcome.